# VULCAN: CRAFTING COMPACT CLASS-SPECIFIC VISION TRANSFORMERS FOR EDGE INTELLIGENCE

**Ziteng Wei**[1], **Qiang He**[1,4*], **Feifei Chen**[2], **Ranjie Duan**[3], **Xiaodan Li**[3],
**Bin Li**[3], **Yuefeng Chen**[3], **Hui Xue**[3], **Hai Jin**[1], **Yun Yang**[4]

[1] National Engineering Research Center for Big Data Technology and System,
Services Computing Technology and System Lab, Cluster and Grid Computing Lab,
Huazhong University of Science and Technology
[2] Deakin University     [3] Alibaba Group     [4] Swinburne University of Technology
`{weiziteng,hqiang,hjin}@hust.edu.cn`

## ABSTRACT

Large Vision Transformers (ViTs) must often be compressed before they can be deployed on resource-constrained edge devices. However, many edge devices require only part of the *all-classes* knowledge of a pre-trained ViT in their corresponding application scenarios. This is overlooked by existing compression methods. Lightweight models produced by these methods retain a substantial amount of class-irrelevant knowledge and suffer suboptimal performance on target classes. To address this, we analyze the knowledge distribution of ViT and reveal a knowledge disentanglement within it: neurons in the feed-forward network (FFN) modules encode class-specific knowledge, while the multi-head attention (MHA) modules capture class-agnostic patterns. Building on this insight, we introduce Vulcan, a pruning-oriented post-training method for deriving compact class-specific models from a pre-trained ViT under given resource budgets. Vulcan follows a novel *train-then-prune* paradigm, which introduces redundancy into ViTs deliberately by collapsing FFN neurons onto those with the highest class-specific activations and by enforcing low-rankness in MHA weights. This design mitigates the irreversible knowledge loss of direct pruning, so that the post-trained model can be compressed into a compact one with negligible performance loss. Notably, the derived edge ViTs not only achieve significant reductions in size and computation but also even surpass the original ViTs in performance on specific classes. Comprehensive experiments with five base ViTs covering three representative visual tasks on four datasets demonstrate that Vulcan-derived ViTs outperform the base ViTs on class-specific tasks by up to 15.12% in accuracy, with only 20%–40% of their sizes. Compared with state-of-the-art structured pruning methods, Vulcan improves class-specific accuracy by up to 13.92%. Code is available at *Vulcan*.

## 1 INTRODUCTION

Vision Transformers (ViTs) have achieved remarkable success in diverse visual tasks, including image recognition (Lu et al., 2025; Fixelle, 2025), object detection (Singh, 2023; Wang et al., 2025a), and instance segmentation (Yang et al., 2022; Ravi et al., 2025). Recent advances largely come from scaling up ViTs, which enhances their representation and generalization ability (Wang et al., 2024; 2025b; Han et al., 2025). While such scaling trends have pushed performance boundaries on various benchmarks, they inevitably result in ViTs that are computationally expensive and memory-intensive (Papa et al., 2024; Saha & Xu, 2025). As a result, these oversized ViTs are typically deployed on cloud servers with sufficient computing resources (Jiang et al., 2025).

As illustrated in Figure 1, cloud deployment often fails to guarantee real-time performance, security, and reliability, while edge deployment can address these issues via local inference (Wen et al., 2023; Liao et al., 2024; Ding et al., 2024; Bonazzi et al., 2025). This highlights the urgent need to unlock the potential of ViTs on edge devices such as drones and autonomous vehicles through model

---

*Corresponding Author.

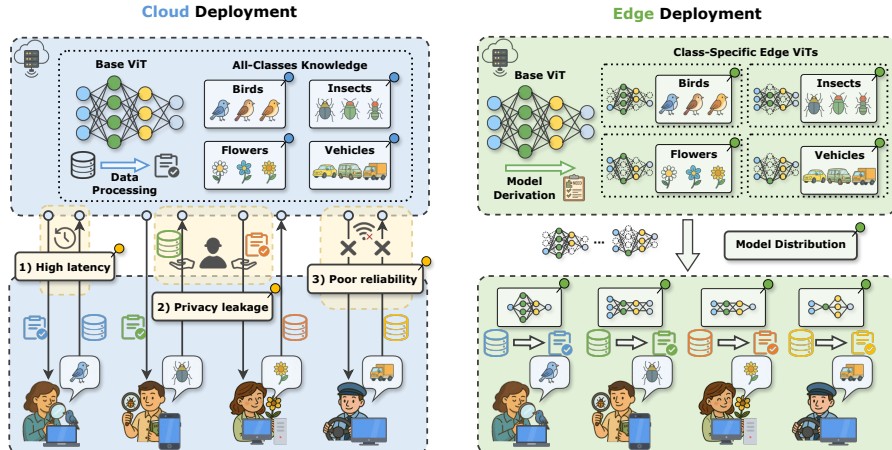

Figure 1: Comparison between cloud and edge ViTs deployment. 1) *Cloud Deployment*: users access models via cloud APIs, suffering from high latency, privacy risks, and poor reliability due to communication and network dependence. 2) *Edge Deployment*: local inference reduces latency, preserves privacy, and improves reliability. Class-specific models are provided for users.

compression (Tuli & Jha, 2023; Ye et al., 2024). However, existing compression methods ignore the fact that edge ViTs, i.e., ViTs deployed on edge devices, typically require only *class-specific* knowledge in their own application scenarios, rather than the *all-classes* knowledge embedded in large-scale pre-trained ViTs (Yao & Abdelzaher, 2023; Zhuang et al., 2024). For example, an in-vehicle sensor tasked with recognizing traffic-related classes such as vehicles, street signs, and traffic lights does not require knowledge about flowers or insects. The presence of irrelevant knowledge distracts a model from focusing on target classes, leading to suboptimal performance. This raises a key question: *how to derive compact class-specific edge ViTs from a general-purpose pre-trained base ViT for lightweight deployment*? (Its necessity is discussed in Appendix A.)

To address this question, we adopt structured pruning (Cheng et al., 2024), an edge-friendly technique (§2), for lightweight deployment. However, existing pruning methods (Zhang et al., 2024; Sun et al., 2025) lack pruning strategies tailored to class specificity. Simply replacing calibration datasets with class-specific data during pruning and retraining is insufficient, as it results in models that still fail to focus on target classes. Moreover, these methods follow the conventional prune-then-train paradigm, which often incurs irreversible knowledge loss, particularly at high pruning rates, since pruned weights may be unimportant but not dispensable. More fundamentally, achieving class-specific model derivation requires an understanding of how class-specific knowledge is distributed across ViT modules—a question that remains largely unresolved, as existing interpretability studies offer only limited insights (Choi et al., 2024; Li et al., 2025).

We investigated the knowledge distribution within ViTs (Geva et al., 2021; Dai et al., 2022) and found a disentangled distribution: FFNs primarily encode interpretable class-specific knowledge, while MHAs capture class-agnostic patterns. Building on this insight, this paper presents Vulcan, a pruning-oriented post-training method that can derive compact class-specific ViTs from a pre-trained ViT. Unlike conventional pruning methods, Vulcan follows a novel *train-then-prune* paradigm that ensures near-lossless pruning after post-training and minimizes knowledge loss during model compression. Specifically, Vulcan employs class-centric neuron collapse to aggregate FFN neurons onto those with the highest activations, deliberately introducing redundancy while enabling class-relevant neurons to dominate feature extraction. Meanwhile, Vulcan applies truncated nuclear-norm regularization to enhance the low-rankness of projection matrices in MHA, enabling near-lossless pruning via singular value decomposition (SVD). Given a specified resource budget (e.g., #Param, GFLOPs), Vulcan integrates these two strategies under an augmented Lagrangian framework (Birgin & Martínez, 2014) to derive edge ViTs. We summarize the key contributions of Vulcan as follows:

- We provide fundamental insights into the disentangled knowledge distribution of ViTs, where FFNs encode class-specific knowledge while MHAs capture class-agnostic patterns.

- To the best of our knowledge, Vulcan is the first technique for deriving compact class-specific ViTs. As a pruning-oriented post-training method, it is also the first to introduce the novel train-then-prune compression paradigm, which minimizes knowledge loss during pruning.

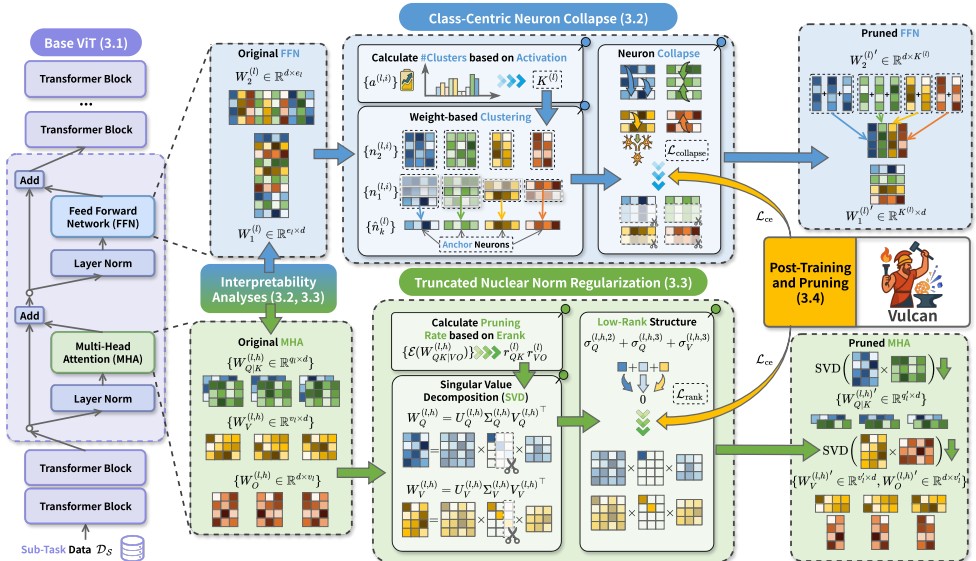

Figure 2: Overview of Vulcan. 1) *Class-Centric Neuron Collapse* (CCNC): neurons in FFN modules are clustered, and all neurons within a cluster collapse into the one with the highest activation for the target classes. 2) *Truncated Nuclear Norm Regularization* (TNNR): low-rank structures are introduced into matrices in MHA modules to support near-lossless SVD-based compression.

- Extensive experiments with five base ViTs covering three typical visual tasks and four benchmarks demonstrate that Vulcan-derived edge ViTs achieve significant reductions in model size while outperforming both the base ViTs and models derived by state-of-the-art structured pruning methods in class-specific performance.

## 2  BACKGROUND AND RELATED WORK

**Edge Model Deployment**. Recently, a series of increasingly large ViTs have been developed (Zhai et al., 2022; Dehghani et al., 2023; Wang et al., 2024), contrasting with the growing demand for deploying models on edge devices for real-time responsiveness, privacy preservation, and reliable service. To address this, several edge-friendly architectures have been proposed, such as Mobile-ViT (Mehta & Rastegari, 2022), EfficientViT (Liu et al., 2023), and Flatten Transformer (Han et al., 2023). While effective, these approaches rely on manual architecture design and require training from scratch. Moreover, such specialized architectures do not naturally scale with the rapid advances of large ViTs, limiting their ability to deliver increasingly powerful models for edge deployment. In contrast, Vulcan focuses on deriving compact ViTs from a pre-trained base ViT, allowing edge ViTs to inherit knowledge from the base ViT and enabling efficient model development.

**Model Compression**. Various compression methods enable edge deployment of ViTs, including quantization (Choi & Kim, 2025; Zhong et al., 2025), knowledge distillation (Yang et al., 2024b; Cao et al., 2025), as well as structured (Zhang et al., 2024; Sun et al., 2025) and unstructured pruning (Chen et al., 2021; Liao et al., 2023). However, not all are edge-friendly (Appendix B). Quantization and unstructured pruning typically rely on specialized infrastructure for acceleration, which limits their applicability on diverse edge devices (Yang et al., 2024a; Cheng et al., 2024). Knowledge distillation transfers knowledge only from the feature space, incurring substantial training cost (He et al., 2025). In contrast, structured pruning extracts knowledge from the parameter space and yields regularly shaped models that can be deployed across diverse edge devices, which serves as the foundation of Vulcan. Different from the conventional prune-then-train paradigm, Vulcan follows a *train-then-prune* paradigm that avoids irreversible knowledge loss caused by direct weight removal and achieves near-lossless pruning. It is worth noting that Vulcan is orthogonal to other compression techniques and can be seamlessly combined with them for even lighter deployment (Appendix J.9).

**Interpretability of ViTs**. Extensive research has investigated the interpretability of ViTs from different perspectives, such as attention visualization (Chefer et al., 2021; Choi et al., 2024), decision pathways (Komorowski et al., 2023; Brinkmann et al., 2024), and robustness to perturbations (Fu et al., 2022; Hu et al., 2024). However, the distribution of *class-specific* knowledge across different

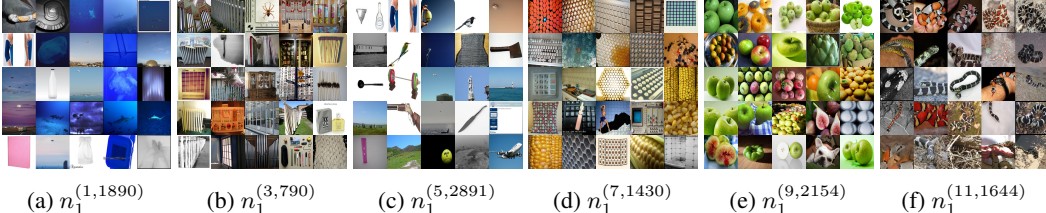

(a) $n_1^{(1,1890)}$  (b) $n_1^{(3,790)}$  (c) $n_1^{(5,2891)}$  (d) $n_1^{(7,1430)}$  (e) $n_1^{(9,2154)}$  (f) $n_1^{(11,1644)}$

Figure 3: Top-25 activated images for random FFN neurons in DeiT-Base. (a)-(f) correspond to patterns of cold color tones, vertical stripes, monotone backgrounds, grid textures, stacked objects, and snakes, illustrating the strong interpretability of FFN neurons. See Appendix C for more examples.

modules of ViTs remains largely unexplored. This missing insight is particularly critical for supporting class-specific model derivation, which is essential to meet the customized requirements of edge users. In this paper, we shed light on this issue by providing an analysis of how class-specific and class-agnostic knowledge are structurally disentangled within ViTs (§3.2-§3.3).

## 3 VULCAN: CLASS-SPECIFIC MODEL DERIVATION

This section introduces Vulcan, a pruning-oriented post-training method for class-specific model derivation. We begin with notations and preliminaries (§3.1), then detail two key components: class-centric neuron collapse for FFNs (§3.2) and truncated nuclear norm regularization for MHAs (§3.3). Finally, we introduce the post-training procedure and the pruning strategy for constructing compact models from the post-trained base ViT (§3.4). An overview of Vulcan is illustrated in Figure 2.

### 3.1 NOTATIONS AND PRELIMINARIES

**Vision Transformer**. A ViT (Dosovitskiy, 2021) consists of a stack of Transformer blocks, each containing an MHA and FFN module. Given a patch token sequence $\mathbf{X} \in \mathbb{R}^{N \times d}$ with $N$ tokens and embedding dimension $d$, each MHA head computes and aggregates contextualized representations:

$$\text{MHA}^{(l)}(\mathbf{X}) = \sum_{h=1}^{H_l} \text{Attn}^{(l,h)}(\mathbf{X}) = \sum_{h=1}^{H_l} \text{softmax}\Big(\frac{\mathbf{X}W_Q^{(l,h)^\top} W_K^{(l,h)} \mathbf{X}^\top}{\sqrt{q_l}}\Big)\mathbf{X}W_V^{(l,h)^\top} W_O^{(l,h)^\top} \quad (1)$$

where $W_{Q|K}^{(l,h)} \in \mathbb{R}^{q_l \times d}$, $W_V^{(l,h)} \in \mathbb{R}^{v_l \times d}$ and $W_O^{(l,h)} \in \mathbb{R}^{d \times v_l}$ are the query, key, value, and output projection matrices for the $h$-th head in the $l$-th block. For simplicity, bias terms are omitted. Following MHA, the FFN module applies a two-layer multi-layer perceptron:

$$\text{FFN}^{(l)}(\mathbf{X}) = \sigma(\mathbf{X}W_1^{(l)^\top})W_2^{(l)^\top} = \sum_{i=1}^{e_l} \sigma(\mathbf{X}W_1^{(l)}[i]) \otimes W_2^{(l)^\top}[i] = \sum_{i=1}^{e_l} \sigma(\mathbf{X}n_1^{(l,i)}) \otimes n_2^{(l,i)} \quad (2)$$

where $W_1^{(l)} \in \mathbb{R}^{e_l \times d}$ and $W_2^{(l)} \in \mathbb{R}^{d \times e_l}$ are the FFN projection matrices in $l$-th block, $\sigma(\cdot)$ is a nonlinear activation (e.g., GELU), $n_{1|2}^{(l,i)} \in \mathbb{R}^d$ is the $i$-th neuron in $W_{1|2}^{(l)}$, and $\otimes$ is the outer product.

**Sub-Task**. Let $\mathcal{M}_B$ be the pre-trained base ViT trained on $\mathcal{D}_\mathcal{Y}$ for classes $\mathcal{Y} = \{y_1, \ldots, y_{|\mathcal{Y}|}\}$. Vulcan aims to derive a compact class-specific edge ViT $\mathcal{M}_E$ from $\mathcal{M}_B$, specialized for a subset $\mathcal{S} \subseteq \mathcal{Y}$. We refer to $\mathcal{S}$ as a *sub-task* of $\mathcal{M}_B$, and denote its corresponding dataset by $\mathcal{D}_\mathcal{S}$.

### 3.2 CLASS-CENTRIC NEURON COLLAPSE FOR FFNS

**Insight**. As shown in Eq. (2), an FFN computes via weighted aggregation, where the activation of each neuron $n_1^{(l,i)}$ serves as weights and $n_2^{(l,i)}$ as values. Thus, activation magnitudes decisively determine neuron contributions to FFN outputs. Inspired by the concept of "knowledge neurons" (Geva et al., 2021; Dai et al., 2022), we conduct an activation-driven analysis on DeiT-Base (Touvron et al., 2021). Specifically, for each neuron $n_1^{(l,i)}$, we sum its activations over all patch tokens for each image in the ImageNet-1K validation set (Russakovsky et al., 2015), and visualize the top-25 images with the highest activations to inspect what kind of knowledge is encoded in the neuron. Figure 3 shows that these neurons exhibit remarkably strong human-recognized interpretability, with knowledge becoming more semantic in deeper blocks. For example, shallow neurons capture simple patterns such as color tones, textures, or backgrounds, while deeper neurons specialize in semantic concepts such as specific classes like snakes. This suggests that FFN modules serve as reservoirs of class-specific knowledge, since different classes can be distinguished by shallow and semantic patterns.

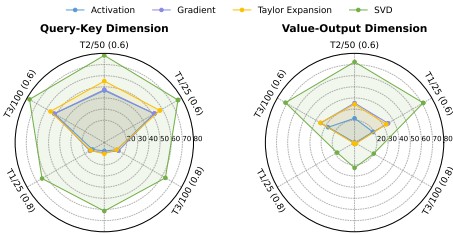
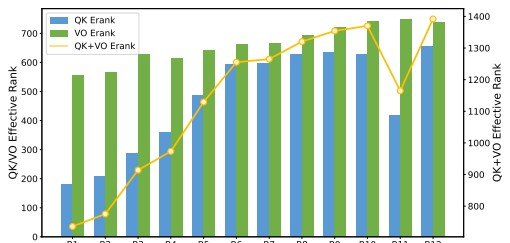

Figure 4: Comparison of accuracy between SVD-based and score-based pruning for DeiT-Base on ImageNet-1K. Tj/$N(R)$ represents sub-task Tj (|Tj|=$N$) with pruning rate $R$. SVD outperforms other methods by a significant margin.

Figure 5: Effective rank (Erank) distribution of QK and VO dimensions across different blocks of DeiT-Base. The QK/VO Erank represents the sum of the Erank values across all heads in these two dimensions for each block.

**Class-Centric Neuron Collapse (CCNC).** Building on this insight, we propose CCNC to introduce redundancy into the FFN intermediate dimension ($e$) for pruning. Given a sub-task $\mathcal{S}$, Vulcan use $\mathcal{D}_\mathcal{S}$ to compute the activations $a^{(l,i)} = \sum_j \sigma(\mathbf{X}[j] \cdot n_1^{(l,i)})$. Higher activation implies stronger relevance of $n_1^{(l,i)}$ to target classes in $\mathcal{S}$. To guide the model to focus on class-specific knowledge, Vulcan performs k-means clustering on $n_1^{(l,i)}$ in each block based on their weights, and collapses all neurons within the same cluster $\mathcal{C}_k^{(l)}$ into the neuron with the highest activation $\hat{a}_k^{(l)}$, termed the *anchor neuron* $\hat{n}_k^{(l)}$. As Figure 3 shows that neurons in different blocks specialize in distinct functions, Vulcan avoids uniform pruning rates across all blocks (Appendix J.2). Instead, it adaptively determines the number of clusters per block based on activation distribution and overall pruning rate $R \in (0,1)$. Formally, the number of clusters in the FFN of the $l$-th block $K^{(l)}$ is calculated as follows:

$$K^{(l)} = \sum_{i=1}^{e_l} \mathbb{I}\bigg(a^{(l,i)} > \Phi\big(\mathcal{A}^{(l)}, \lceil (\sum_{j=1}^{L} e_j) \times R \rceil\big)\bigg), \quad \mathcal{A}^{(l)} = \bigcup_{l=1}^{L}\{a^{(l,i)} | i = 1, \ldots, e_l\} \quad (3)$$

where $\Phi(\mathcal{A}, k)$ returns the $k$-th smallest element in set $\mathcal{A}$. To realize cluster-level collapse, Vulcan enforces the *weights or activations* of all neurons within the same cluster to contract toward the anchor neuron. This is achieved by introducing a collapse regularization term during post-training:

$$\mathcal{L}_{\text{collapse}} = \sum_{l=1}^{L} \frac{1}{K^{(l)}} \sum_{k=1}^{K^{(l)}} \sum_{i=1}^{|\mathcal{C}_k^{(l)}|} |\nu_k^{(l,i)} - \hat{\nu}_k^{(l)}|, \quad \hat{\nu}_k^{(l)} = \nu_k^{(l,i^*)}, \; i^* = \underset{i \in [|\mathcal{C}_k^{(l)}|]}{\arg\max} \, a_k^{(l,i)} \quad (4)$$

where $\mathcal{C}_k^{(l)}$ is the $k$-th cluster in the $l$-th block, and $\nu_k^{(l,i)}$ denotes either the weight $n_k^{(l,i)}$ or the activation $a_k^{(l,i)}$ of the $i$-th neuron in $\mathcal{C}_k^{(l)}$, with a unique choice applied in practice. Experiments show that weight collapse converges faster and performs better than activation collapse (Figure 10). Through Eq. (4), Vulcan guides the model to focus on class-specific knowledge and enables each cluster to be represented by a single anchor neuron, which achieves effective compression of the FFN modules. Appendix D further shows that CCNC can also be applied to other FFN architectures.

### 3.3 TRUNCATED NUCLEAR NORM REGULARIZATION FOR MHAS

**Insight.** For MHA, Vulcan prunes the query-key (QK: $q$) and value-output dimensions (VO: $v$), enforcing identical sizes across heads in each block (Appendix E). As shown in Eq. (1), the QK and VO dimensions are intermediate dimensions of two matrix multiplications (i.e., $W_Q^\top W_K, W_V^\top W_O^\top \in \mathbb{R}^{d \times d}$), indicating that they can be pruned through singular value decomposition (SVD). Surprisingly, we observe that this data-free SVD-based method consistently outperforms data-independent score-based pruning methods (Yu et al., 2018; Chen et al., 2021; Yang et al., 2023), as shown in Figure 4. It suggests that effective compression of MHA can be achieved even when the sub-task is unknown. This method-driven analysis indicates that the QK and VO dimensions in MHA store class-agnostic knowledge. We provide a theoretical explanation for this observation in Appendix F.

**Truncated Nuclear Norm Regularization (TNNR).** To leverage SVD for near-lossless MHA pruning, we propose TNNR to introduce low-rank structures into $W_Q^{(l,h)}$ and $W_V^{(l,h)}$, with their low-rank properties shared by $W_K^{(l,h)}$ and $W_O^{(l,h)}$, respectively. We use effective rank (Roy & Vetterli, 2007) to measure the knowledge capacity of each attention head and observe that the knowledge in MHA is also unevenly distributed across different blocks and dimensions, as shown in Figure 5. Based on this observation, Vulcan adaptively determines the pruning rate $R_{QK|VO}^{(l)}$ for the QK and VO

dimensions according to the dimension-level effective ranks and the overall pruning rate $R$:

$$R_{QK|VO}^{(l)} = \frac{2r_{VO|QK}^{(l)}}{r_{QK}^{(l)} + r_{VO}^{(l)}} R^{(l)}, \quad R^{(l)} = \frac{L}{\sum_{i=1}^{L} 1/(r_{QK}^{(i)} + r_{VO}^{(i)})} \cdot \frac{R}{r_{QK}^{(l)} + r_{VO}^{(l)}} \tag{5}$$

$$r_{QK|VO}^{(l)} = \sum_{h=1}^{H_l} \mathcal{E}(W_{QK|VO}^{(l,h)}), \quad \mathcal{E}(W) = \exp(-\sum_i p_i \log p_i), \quad p_i = \frac{\sigma_i}{\sum_j \sigma_j} \tag{6}$$

where $W_{QK}^{(l,h)} = W_Q^{(l,h)\top} W_K^{(l,h)}$, $W_{VO}^{(l,h)} = W_V^{(l,h)\top} W_O^{(l,h)\top}$, $\mathcal{E}(W)$ is the effective rank of matrix $W$ and $\sigma_i$ is the $i$-th singular value of $W$. Then, Vulcan applies truncation to the nuclear norm of $W_Q^{(l,h)}$ and $W_V^{(l,h)}$ according to Eq. (5) and constructs a regularization term:

$$\mathcal{L}_{\text{rank}} = \sum_{l=1}^{L} \sum_{h=1}^{H_l} \left( \sum_{i=q_l'+1}^{q_l} \sigma_Q^{(l,h,i)} + \sum_{i=v_l'+1}^{v_l} \sigma_V^{(l,h,i)} \right), q_l' = \lfloor q_l(1-R_{QK}^{(l)}) \rfloor, v_l' = \lfloor v_l(1-R_{VO}^{(l)}) \rfloor \tag{7}$$

where $\sigma_{Q|V}^{(l,h,i)}$ is the $i$-th singular value of $W_{Q|V}^{(l,h)}$. With Eq. (7), Vulcan effectively extracts class-agnostic knowledge from the MHA and supports near-lossless pruning after post-training.

## 3.4 POST-TRAINING AND PRUNING

**Objective**. Vulcan post-training aims to focus the model on specific classes while introducing redundancy in the three ViT dimensions ($e$, $q$, $v$) for pruning. Eqs. (4) and (7) serve as redundancy constraints that guide the compression process. To strictly enforce these constraints during post-training, Vulcan uses the augmented Lagrangian framework to construct the final objective function. Specifically, it extends Eqs. (4) and (7) into linear and quadratic functions, and introduces two learnable Lagrange multipliers $\lambda_1$ and $\lambda_2$ to control convergence. The final loss function is as follows:

$$\mathcal{L} = \mathcal{L}_{\text{T}} + \sum_{l=1}^{L} \frac{1}{K^{(l)}} \sum_{k=1}^{K^{(l)}} \sum_{i=1}^{|\mathcal{C}_k^{(l)}|} \left( \lambda_1 |\nu_k^{(l,i)} - \hat{\nu}_k^{(l)}| + \lambda_2 (\nu_k^{(l,i)} - \hat{\nu}_k^{(l)})^2 \right)$$
$$+ \sum_{l=1}^{L} \sum_{h=1}^{H_l} \left( \sum_{i=q_l'+1}^{q_l} \left( \lambda_1 \sigma_Q^{(l,h,i)} + \lambda_2 \sigma_Q^{(l,h,i)2} \right) + \sum_{i=v_l'+1}^{v_l} \left( \lambda_1 \sigma_V^{(l,h,i)} + \lambda_2 \sigma_V^{(l,h,i)2} \right) \right) \tag{8}$$

where $\mathcal{L}_{\text{T}}$ is the loss term for visual tasks, as well as $\lambda_1$ and $\lambda_2$ are initialized to zero and updated using gradient ascent with a penalty parameter $\rho$ as the learning rate.

**Pruning**. After post-training, Vulcan can directly derive a compact class-specific edge ViT from the post-trained base ViT with negligible performance loss (Section 3.5). Specifically, the pruned FFN module contains two new weights, $W_1^{(l)'} \in \mathbb{R}^{K^{(l)} \times d}$ and $W_2^{(l)'} \in \mathbb{R}^{d \times K^{(l)}}$, which satisfy:

$$W_1^{(l)'}[k] = \hat{n}_k^{(l)}, \quad W_2^{(l)'}[:,k] = \sum_{i=1}^{|\mathcal{C}_k^{(l)}|} n_{k,2}^{(l,i)}, \quad k \in [K^{(l)}] \tag{9}$$

where $n_{k,2}^{(l,i)}$ denotes the neuron in $W_2^{(l)}$ corresponding to $n_k^{(l,i)}$. This process is essentially the inverse of network expansion (Ding et al., 2023; Yao et al., 2024) and adheres to the function-preserving principle (Chen et al., 2016). For the pruned MHA module, its four new weights, $W_{Q|K}^{(l,h)'} \in \mathbb{R}^{q_l' \times d}$, $W_V^{(l,h)'} \in \mathbb{R}^{v_l' \times d}$, and $W_O^{(l,h)'} \in \mathbb{R}^{d \times v_l'}$ can be directly obtained through SVD:

$$W_Q^{(l,h)'} = (U_{QK}^{(l,h)}[:,:q_l'] \Sigma_{QK}^{(l,h)}[:q_l',:q_l'])^\top \times \sqrt{q_l'/q_l}, \quad W_K^{(l,h)'} = (V_{QK}^{(l,h)}[:,:q_l'])^\top \tag{10}$$

$$W_V^{(l,h)'} = (U_{VO}^{(l,h)}[:,:v_l'] \Sigma_{VO}^{(l,h)}[:v_l',:v_l'])^\top, \quad W_O^{(l,h)'} = V_{VO}^{(l,h)}[:,:v_l'] \tag{11}$$

where $U_{QK}^{(l,h)} \Sigma_{QK}^{(l,h)} V_{QK}^{(l,h)\top} = W_{QK}^{(l,h)}$ and $U_{VO}^{(l,h)} \Sigma_{VO}^{(l,h)} V_{VO}^{(l,h)\top} = W_{VO}^{(l,h)}$. It is worth noting that conventional pruning methods remove weights based on importance scores, which may discard useful knowledge. In contrast, Vulcan ensures that the pruned weights are entirely redundant and contain no valuable knowledge. Pseudocode and notation tables are provided in Appendix K and L.

### 3.5 PROOF OF LOSSLESS PERFORMANCE AFTER PRUNING

As discussed in Section 3.4, once the constraints in Eq. (4) and Eq. (7) are fully satisfied, i.e., $\mathcal{L}_{\text{collapse}} = 0$ and $\mathcal{L}_{\text{rank}} = 0$, Vulcan ensures lossless pruning for the post-trained model. In this section, we provide a formal proof of this property.

***Proof1: Lossless Performance after Pruning***. After post-training, the base ViT satisfies the following two key conditions:

1. All neurons within the same cluster $\mathcal{C}_k^{(l)}$ in the FFN module in $l$-th block collapse to the anchor neuron $\hat{n}_k^{(l)}$ of that cluster.

2. For the $h$-th attention head in $l$-th block, the matrices $W_Q^{(l,h)}$ and $W_V^{(l,h)}$ contain exactly $q_l'$ and $v_l'$ non-zero singular values, respectively. This means $\text{rank}(W_Q^{(l,h)}) \le q_l'$ and $\text{rank}(W_V^{(l,h)}) \le v_l'$.

**FFN Case**. According to Eq. (2), the output of the FFN in the $l$-th block can be expressed as:

$$\begin{aligned}
\text{FFN}^{(l)}(\mathbf{X}) &= \sum_{i=1}^{e_l} \sigma(\mathbf{X} n_1^{(l,i)}) \otimes n_2^{(l,i)} = \sum_{k=1}^{K^{(l)}} \sum_{j=1}^{|\mathcal{C}_k^{(l)}|} \sigma(\mathbf{X} n_k^{(l,j)}) \otimes n_{k,2}^{(l,j)} \\
&= \sum_{k=1}^{K^{(l)}} \sum_{j=1}^{|\mathcal{C}_k^{(l)}|} \sigma(\mathbf{X} \hat{n}_k^{(l)}) \otimes n_{k,2}^{(l,j)} = \sum_{k=1}^{K^{(l)}} \sigma(\mathbf{X} \hat{n}_k^{(l)}) \otimes \sum_{j=1}^{|\mathcal{C}_k^{(l)}|} n_{k,2}^{(l,j)}
\end{aligned} \quad (12)$$

where $n_{k,2}^{(l,j)}$ denotes the neuron in $W_2^{(l)}$ corresponding to $n_k^{(l,j)}$. According to Eq. (9), the output of the pruned FFN in the $l$-th block can be expressed as:

$$\begin{aligned}
\text{FFN}_{\text{pruned}}^{(l)}(\mathbf{X}) &= \sum_{k=1}^{e_l'} \sigma(\mathbf{X} W_1^{(l)'}[k]) \otimes {W_2^{(l)'}}^\top[k] = \sum_{k=1}^{K^{(l)}} \sigma(\mathbf{X} W_1^{(l)'}[k]) \otimes W_2^{(l)'}[:,k] \\
&= \sum_{k=1}^{K^{(l)}} \sigma(\mathbf{X} \hat{n}_k^{(l)}) \otimes \sum_{j=1}^{|\mathcal{C}_k^{(l)}|} n_{k,2}^{(l,j)} = \text{FFN}^{(l)}(\mathbf{X})
\end{aligned} \quad (13)$$

**MHA Case**. According to the second key condition, we can conclude that:

$$\begin{aligned}
\text{rank}({W_Q^{(l,h)}}^\top W_K^{(l,h)}) &\le \min(\text{rank}(W_Q^{(l,h)}), \text{rank}(W_K^{(l,h)})) \le q_l' \\
\text{rank}({W_V^{(l,h)}}^\top W_O^{(l,h)}) &\le \min(\text{rank}(W_V^{(l,h)}), \text{rank}(W_O^{(l,h)})) \le v_l'
\end{aligned} \quad (14)$$

Therefore, the pruning process based on SVD is lossless, i.e., the pruned matrices ${W_{Q|K|V|O}^{(l,h)}}'$ satisfy:

$$ {W_Q^{(l,h)'}}^\top W_K^{(l,h)'} = {W_Q^{(l,h)}}^\top W_K^{(l,h)}, {W_V^{(l,h)'}}^\top {W_O^{(l,h)'}}^\top = {W_V^{(l,h)}}^\top {W_O^{(l,h)}}^\top \quad (15)$$

Substituting these equalities into the Eq. (1), we obtain:

$$\begin{aligned}
\text{MHA}_{\text{pruned}}^{(l)}(\mathbf{X}) &= \sum_{h=1}^{H_l} \text{softmax}\left( \frac{\mathbf{X} {W_Q^{(l,h)'}}^\top W_K^{(l,h)'} \mathbf{X}^\top}{\sqrt{q_l}} \right) \mathbf{X} {W_V^{(l,h)'}}^\top {W_O^{(l,h)'}}^\top \\
&= \sum_{h=1}^{H_l} \text{softmax}\left( \frac{\mathbf{X} {W_Q^{(l,h)}}^\top W_K^{(l,h)} \mathbf{X}^\top}{\sqrt{q_l}} \right) \mathbf{X} {W_V^{(l,h)}}^\top {W_O^{(l,h)}}^\top = \text{MHA}^{(l)}(\mathbf{X})
\end{aligned} \quad (16)$$

Thus, both the FFN and MHA modules of the pruned model produce identical outputs to those of the post-trained base ViT, which completes the proof. □

In practice, since $\mathcal{L}_{\text{collapse}}$ and $\mathcal{L}_{\text{rank}}$ only asymptotically approach zero, model accuracy before and after pruning exhibits minimal fluctuations, indicating that Vulcan achieves near-lossless pruning.

## 4 EXPERIMENTS AND ANALYSIS

**Models and Datasets**. We evaluate Vulcan on two widely adopted ViT families: DeiT-Base/Small/Tiny (Touvron et al., 2021) and Fast/Mask R-CNN (Swin-T) (Liu et al., 2021). For recognition tasks, we use ImageNet-1K (Russakovsky et al., 2015) and CIFAR-100/10 (Krizhevsky

Table 1: Overall performance of Vulcan and baselines on DeiT-Base with ImageNet-1K sub-tasks of different sizes (25, 50, 100 classes) under pruning rates of 0.60 and 0.80. The best top-1 accuracy (%) in each column is highlighted in bold, while the second best is underlined. '(FT)' denotes fine-tuning on the class-specific data. Results for other pruning rates are provided in Appendix J.1.

| Method | T1/25 | T2/25 | T3/25 | *Avg* | T4/50 | T5/50 | T6/50 | *Avg* | T7/100 | T8/100 | T9/100 | *Avg* |
|---|---|---|---|---|---|---|---|---|---|---|---|---|
| **DeiT-Base** | 79.92 | 82.56 | 80.67 | 81.05 | 80.48 | 79.64 | 84.96 | 81.69 | 80.49 | 84.12 | 78.58 | 81.06 |
| **DeiT-Base (FT)** | 96.40 | 96.96 | 96.96 | 96.77 | 95.00 | 93.36 | 94.68 | 94.35 | 91.56 | 93.29 | 92.84 | 92.56 |
| *Pruning Rate = 0.60* | | | | | | | | | | | | |
| **Random** | 91.76 | 92.72 | 89.69 | 91.39 | 86.16 | 85.60 | 87.56 | 86.44 | 83.11 | 86.10 | 80.42 | 83.21 |
| **NViT** | 83.91 | 87.12 | 85.62 | 85.55 | 81.64 | 81.20 | 80.72 | 81.19 | 78.31 | 80.31 | 78.12 | 78.91 |
| **X-Pruner** | 91.68 | 93.04 | 93.21 | 92.64 | 90.24 | 88.28 | 90.16 | 89.56 | 84.51 | 87.12 | 86.46 | 86.03 |
| **DC-ViT** | 71.60 | 83.28 | 76.92 | 77.27 | 69.48 | 67.20 | 66.96 | 67.88 | 60.06 | 64.98 | 65.18 | 64.41 |
| **MDP** | 91.04 | 94.80 | 92.09 | 92.64 | 90.32 | 85.00 | 91.16 | 88.83 | 82.61 | 84.33 | 82.86 | 83.27 |
| *Vulcan* | 95.04 | 96.24 | 95.53 | **95.60** | 92.16 | 91.72 | 92.44 | **92.11** | 88.03 | 90.70 | 89.02 | **89.25** |
| *Pruning Rate = 0.80* | | | | | | | | | | | | |
| **Random** | 71.36 | 78.48 | 82.90 | 77.58 | 71.92 | 71.16 | 73.28 | 72.12 | 65.28 | 67.71 | 60.98 | 64.66 |
| **NViT** | 64.16 | 58.00 | 60.38 | 60.85 | 52.76 | 53.44 | 49.04 | 51.75 | 41.56 | 47.62 | 48.74 | 45.97 |
| **X-Pruner** | 85.60 | 86.00 | 85.62 | 85.74 | 76.24 | 74.00 | 79.72 | 76.65 | 72.62 | 76.74 | 75.52 | 74.95 |
| **DC-ViT** | 54.72 | 66.24 | 64.46 | 61.81 | 47.52 | 53.40 | 50.84 | 50.59 | 35.66 | 35.12 | 34.66 | 35.15 |
| **MDP** | 83.68 | 88.08 | 84.66 | 85.47 | 74.04 | 74.20 | 79.88 | 76.04 | 61.72 | 67.81 | 67.54 | 65.69 |
| *Vulcan* | 92.24 | 94.32 | 92.57 | **93.04** | 88.36 | 88.12 | 88.52 | **88.33** | 81.82 | 85.20 | 82.84 | **83.29** |

et al., 2009); for detection and segmentation tasks, we use COCO (Lin et al., 2014). Sub-tasks of varying scales are constructed by randomly sampling classes from each dataset.

**Implementations**. GFLOPs (Appendix G) is used as the metric to compute pruning rates. During post-training, we set the batch size to 256, the learning rate to $10^{-4}$, the penalty parameter $\rho$ to 1.0, and use the AdamW for optimization. For CCNC, we apply z-score normalization to the activation values across blocks before computing $K^{(l)}$. Anchor neurons are updated per batch. To ensure that the derived models can achieve accelerated inference on a wide range of edge devices, Vulcan aligns the pruned dimensions to multiples of 8 (NVIDIA, 2023). See Appendix H for more details.

**Baselines**. We compare Vulcan with five state-of-the-art structured pruning methods: Random Pruning (Gadhikar et al., 2023), NViT (Yang et al., 2023), X-Pruner (Yu & Xiang, 2023), DC-ViT (Zhang et al., 2024), and MDP (Sun et al., 2025) (Appendix I). To ensure fairness, we retrain the models pruned by these methods until they converge.

## 4.1 RESULTS

**Class-Agnostic *vs*. Class-Specific**. We compare class-agnostic and class-specific methods to highlight the advantages of the latter. Baselines derive class-agnostic models from DeiT-Base for all ImageNet classes, while Vulcan derives class-specific models tailored to the target sub-task $\mathcal{S}$. Figure 6 shows that Vulcan consistently outperforms baselines in class-specific accuracy at the same pruning rate. This demonstrates Vulcan's advantage in meeting edge devices' customized needs.

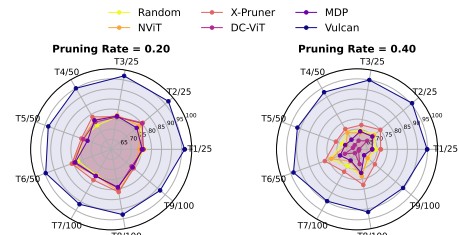

Figure 6: Comparison between class-agnostic and class-specific model derivation. Tj/$N$ represents sub-task Tj with $N$ random classes.

**Overall Performance**. For a fair comparison, we replace the calibration datasets originally used by the baselines with the sub-task dataset $\mathcal{D}_\mathcal{S}$, which adapts them to the setting of class-specific model derivation. As shown in Table 1, Vulcan-derived edge ViTs achieve up to 15.12% higher accuracy than the base ViT and up to 13.92% over the models derived by state-of-the-art baselines across different sub-tasks and pruning rates. At a pruning rate of 0.60, Vulcan improves accuracy by 11.05% over the base ViT, and surpasses the two best-performing baselines, X-Pruner and MDP, by 2.91% and 4.07%, respectively. Even at a high pruning rate of 0.80, where all methods suffer from significant accuracy degradation, Vulcan still delivers strong performance, yielding improvements of 6.95%, 9.11%, and 12.49% over the base ViT, X-Pruner, and MDP, respectively. It can be seen that the advantages of Vulcan become more pronounced as the pruning rate increases and the sub-task size grows. Meanwhile, Vulcan is able to retain 97.63% and 93.30% of the accuracy of the fine-tuned base ViT at pruning rates of 0.60 and 0.80, respectively. These results demonstrate Vulcan's strong capability to specialize models for target classes.

Figure 7: Performance of edge ViTs derived by Vulcan across different base models and datasets.

Table 2: Comparison of computational efficiency between DeiT-Base and edge ViTs derived by Vulcan under different $R$, evaluated on Jetson Orin NX (bz=1) and NVIDIA RTX 4090 (bz=256).

| Methods | Latency (ms) | | Throughput (image/s) | | Memory (GB) | | #Param | #FLOPs |
|---|---|---|---|---|---|---|---|---|
| | Orin NX | RTX 4090 | Orin NX | RTX 4090 | Orin NX | RTX 4090 | (M) | (G) |
| **DeiT-Base** | 45.45 | 274.27 | 22.00 | 933.39 | 0.34 | 2.21 | 86.57 | 17.57 |
| **Vulcan (0.20)** | 36.84 (1.23×) | 218.16 (1.26×) | 27.14 | 1173.43 | 0.27 | 2.14 | 67.22 | 13.56 (↓21.51%) |
| **Vulcan (0.40)** | 29.73 (1.53×) | 176.01 (1.56×) | 33.63 | 1454.42 | 0.21 | 1.82 | 51.00 | 10.24 (↓41.72%) |
| **Vulcan (0.60)** | 21.81 (2.16×) | 136.67 (2.01×) | 45.86 | 1873.11 | 0.15 | 1.66 | 34.09 | 6.77 (↓61.47%) |
| **Vulcan (0.80)** | 15.06 (3.02×) | 99.59 (2.75×) | 66.41 | 2570.61 | 0.08 | 1.40 | 16.96 | 3.26 (↓81.45%) |

**Generality Across Models and Datasets**. To further validate the generality of Vulcan, we extend our evaluation to different base models (DeiT-Small/Tiny and Fast/Mask R-CNN with Swin-T backbone) and datasets (CIFAR-100/10 and COCO2017). As shown in Figure 7, for recognition tasks, the derived edge ViTs consistently outperform the base ViT on sub-tasks when the pruning rate is below 0.60. Even at pruning rates higher than 0.80, the accuracy of edge ViTs does not degrade severely enough to make the models unusable. For some small-scale sub-tasks, edge ViTs even significantly outperform the base ViT at pruning rates as high as 0.90. For detection and segmentation tasks, Vulcan also effectively extracts class-specific (i.e., person, car, and cat) knowledge from Swin Transformer backbones and yields edge ViTs that rival the performance of the base ViT. See Appendix J.4–J.7 for experiments on generalization, robustness, and wall-clock cost.

**Computational Efficiency**. We evaluate the efficiency of edge ViTs derived by Vulcan on both a representative edge device, Jetson Orin NX, and an NVIDIA RTX 4090. Considering that requests typically arrive sequentially in real-time edge scenarios, whereas servers process requests in batches, we use batch sizes of 1 and 256 to measure the speedup on Orin NX and RTX 4090, respectively. All results are averaged over the models for nine sub-tasks listed in Table 1. As shown in Table 2, when the pruning rate ranges from 0.20 to 0.80, Vulcan-derived models achieve 1.23×–3.02× speedup on Orin NX and 1.26×–2.75× on RTX 4090, while reducing memory consumption by 20.59%–76.47% and 3.17%–36.65%, respectively. See Appendix J.3 for comparisons with baselines.

## 4.2 ANALYSIS

We further analyze the effectiveness of Vulcan from multiple perspectives. Unless otherwise specified, all experiments are conducted on DeiT-Base at a pruning rate of 0.60, with the sub-tasks T1–T3 in Table 1 constructed by sampling classes from ImageNet-1K.

**Understanding the Post-Training Process**. We visualize the post-training process of Vulcan by tracking 1) the accuracy of the derived edge ViT (Acc-Pruned) and the base ViT (Acc-Base) as well as their accuracy gap (∆Acc), and 2) the overall loss $\mathcal{L}$ in Eq. (8). As shown in Figure 8, the post-training process can be viewed as a gradual alignment between the edge ViT and the base ViT. At the early stage, the task loss $\mathcal{L}_T$ dominates, leading to an improvement in the accuracy of the base ViT. As the Lagrange multipliers $\lambda_{1,2}$ increase, the redundancy-enforcing losses $\mathcal{L}_{\text{collapse}}$ and $\mathcal{L}_{\text{rank}}$ begin to dominate. Consequently, the accuracy of the base ViT slightly decreases while that of the edge ViT increases substantially. In the later stage, the two models converge, achieving equivalence.

**Penalty Parameter** $\rho$. As discussed in Section 3.4, the penalty parameter $\rho$, as the only hyperparameter of Vulcan, serves as the learning rate for updating $\lambda_{1,2}$. It determines when $\mathcal{L}_{\text{collapse}}$ and $\mathcal{L}_{\text{rank}}$ dominate post-training and align edge ViTs with base ViT. However, Vulcan is largely insensitive to $\rho$. As shown in Figure 9, when $\rho$ is set to 0.1, 1.0, or 10.0, both the convergence speed and the final accuracy of the edge ViT remain almost unchanged, which demonstrates the robustness of Vulcan. We further analyze the effects of batch size and learning rate in Appendix J.8.

**Weight *vs*. Activation Collapse**. As discussed in Section 3.2, the term $\nu_k^{(l,i)}$ in $\mathcal{L}_{\text{collapse}}$ can be defined in two ways: using the weight vector $n_k^{(l,i)}$ (*weight collapse*) or the activation value $a_k^{(l,i)}$

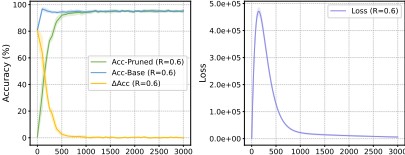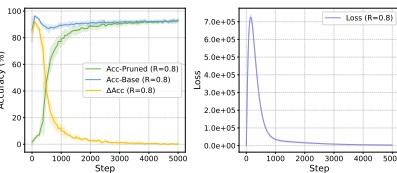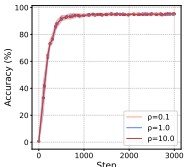

Figure 8: Mean accuracy and loss curves with standard deviation bands on sub-tasks T1/25, T2/25, and T3/25 when deriving models from DeiT-Base using Vulcan at pruning rates 0.60 and 0.80.

Figure 9: Vulcan is insensitive to the choice of $\rho$.

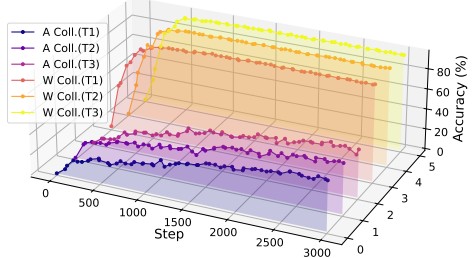

Figure 10: Accuracy trajectories of derived edge ViTs during post-training with weight collapse (W Coll.) and activation collapse (A Coll.).

Table 3: Ablation study of Vulcan. At a pruning rate of 0.60, the edge ViTs derived from DeiT-Base for classes in sub-tasks T1–T3 exhibit significant accuracy degradation when any of CCNC, TNNR, or anchor neurons are removed.

| Setting | T1/25 | T2/25 | T3/25 | Avg |
|---|---|---|---|---|
| **Vulcan** | 95.04 | 96.24 | 95.53 | 95.60 |
| **w/o CCNC** | 8.08 | 13.68 | 12.30 | 11.35 (↓84.25%) |
| **w/o TNNR** | 79.36 | 81.60 | 77.38 | 79.45 (↓16.15%) |
| **w/o anchor** | 90.00 | 92.32 | 90.81 | 91.04 (↓4.56%) |

(*activation collapse*). As shown in Figure 10, weight collapse achieves both faster convergence and higher final accuracy compared to activation collapse. We attribute this to two main reasons: 1) weight collapse directly enforces equality among weights, which better aligns with the pruning process of Vulcan; and 2) the $\mathcal{L}_{\text{collapse}}$ under activation collapse is smaller in magnitude, resulting in weaker constraints. Moreover, we observe that simply increasing $\rho$ does not alleviate this issue.

**Ablation Study**. Class-centric neuron collapse (CCNC) and truncated nuclear norm regularization (TNNR) are Vulcan's core components, responsible for extracting class-relevant knowledge from the FFN and MHA modules, respectively. Table 3 summarizes their effectiveness. Removing CCNC causes an 84.25% accuracy drop due to irreversible knowledge loss in the FFN. Without TNNR, accuracy drops 16.15%, smaller since MHA matrices are inherently low-rank. We further compare collapsing neurons toward the anchor neuron versus a random neuron within each cluster, and find that the absence of anchor guidance leads to an average accuracy drop of 4.56%. These results demonstrate that all designs in Vulcan are indispensable for effective class-specific model derivation.

**Visualization**. To further show that the models derived by Vulcan indeed specialize in the target classes, we derive an edge ViT $\mathcal{M}_{E/dogs}$ from DeiT-Base at a pruning rate of 0.60 for the Stanford Dogs (Khosla et al., 2011), a subset of ImageNet consisting of 120 dog classes. Following the same procedure as Figure 3, we visualize the 680 neurons in the last block of $\mathcal{M}_{E/dogs}$ and identify 229 neurons specialized in recognizing dog-related classes. Representative examples are shown in Figure 11. This demonstrates Vulcan's ability to effectively guide the model toward target classes.

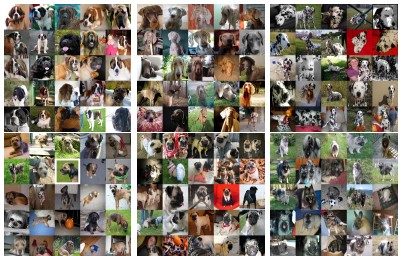

Figure 11: Dog-related neurons in the last block of Vulcan-derived $\mathcal{M}_{E/dogs}$.

## 5 CONCLUSION

This paper presented Vulcan, a novel pruning-oriented post-training method that derives class-specific models from pre-trained ViTs for deployment on edge devices. Motivated by the insight that the FFN modules of ViTs primarily encode class-specific knowledge while the MHA modules capture class-agnostic patterns, Vulcan adopts a novel train-then-prune compression paradigm, leveraging class-centric neuron collapse and truncated nuclear norm regularization to introduce redundancy into the FFN and MHA modules, respectively. This design allows Vulcan to derive compact class-specific edge ViTs from the post-trained base ViTs under resource constraints with negligible performance loss. Extensive experiments demonstrate its effectiveness, generality, and robustness.

## ACKNOWLEDGEMENTS

We sincerely thank the AC and reviewers for their constructive and valuable feedback. This research was supported by the National Key R&D Program of China under Grant No. 2023YFB4502400. It is also supported by the Alibaba Group.

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

## A    NECESSITY OF MODEL DERIVATION

When customizing models for specific edge devices, one may consider directly fine-tuning small models to meet resource and task requirements. However, this approach is often impractical in real-world settings. In contrast, deriving a small model from a large model offers several distinct advantages:

- *Flexibility*. Edge devices exhibit highly diverse resource budgets, and it is unrealistic to assume that a suitably sized pretrained lightweight model always exists for every deployment profile. Training a lightweight model from scratch is also undesirable, as poor initialization often leads to suboptimal convergence, as shown in Table 4. In contrast, deriving small models from a large base model avoids these restrictions. Vulcan can automatically determines the target architecture and transfer useful knowledge directly from the large model into the derived one.
- *Higher Accuracy*. Large pretrained models possess richer representational capacity and encode high-level knowledge that small models do not contain. Models derived through Vulcan inherit this knowledge and therefore achieve substantially better decision boundaries.
- *Scalability*. The paradigm of deriving small models from a large model is forward-compatible: as foundation models continue to improve, the derived models will naturally become stronger as well, without requiring redesign or retraining from scratch.

Table 4: Accuracy comparison on three ImageNet sub-tasks (T1/25–T3/25) between DeiT-Small fine-tuned with and without initialization as well as models of the same size derived from pretrained DeiT-Base using Vulcan.

| Setting | T1/25 | T2/25 | T3/25 | *Avg* |
|---|---|---|---|---|
| **DeiT-Small (w/o init)** | 58.24 | 60.24 | 61.58 | 60.02 |
| **DeiT-Small (w/ init)** | 93.96 | 94.64 | 93.61 | 94.07 |
| **DeiT-Base (Vulcan-0.73)** | **94.80** | **96.08** | **94.65** | **95.18** |

## B    COMPARISON OF MODEL COMPRESSION TECHNIQUES

**Quantization and Unstructured Pruning**. As discussed in Section 2, quantization and unstructured pruning cannot deliver practical acceleration on edge devices without specialized hardware or software support. To empirically validate this point, we conduct experiments on Jetson Orin NX, a widely used embedded device in robotics, industrial inspection, and autonomous driving. Specifically, we apply INT8 quantization (Xi et al., 2024) and magnitude pruning (Han et al., 2015) as representative techniques of quantization and unstructured pruning, respectively, on DeiT-Base, DeiT-Small, and DeiT-Tiny (Touvron et al., 2021). We then measure inference latency with a batch size of 1, since edge devices are typically deployed in real-time scenarios where inputs arrive sequentially. As shown in Table 5, INT8-quantized models cannot be executed on the GPU of Jetson Orin NX due to the lack of INT8 support, and are instead forced to run on the CPU, which results in significantly slower inference compared to the original base ViTs. For magnitude pruning, although the pruned models can still run on the GPU, the parameter count remains unchanged, leading to virtually no difference in inference latency before and after pruning. This result suggests that relying on quantization or unstructured pruning would undermine the latency advantage of edge deployment.

Table 5: Inference latency and throughput of ViTs before and after INT8 quantization and magnitude pruning. Quantized models run on CPU due to lack of TensorRT support, resulting in much slower inference, while magnitude pruning does not reduce parameter count and thus yields no speedup.

| Methods | Base ViT (GPU) | | INT8 Quantization (CPU) | | Magnitude Pruning (GPU) | |
|---|---|---|---|---|---|---|
| | Latency (ms) | Throughput (image/s) | Latency (ms) | Throughput (image/s) | Latency (ms) | Throughput (image/s) |
| **DeiT-Base** | 45.45 | 22.00 | 3429.48 | 0.29 | 45.47 | 21.99 |
| **DeiT-Small** | 15.58 | 64.16 | 1497.11 | 0.67 | 15.80 | 63.28 |
| **DeiT-Tiny** | 14.52 | 68.87 | 666.43 | 1.50 | 14.88 | 67.21 |

**Knowledge Distillation**. As discussed in Section 2, knowledge distillation (KD) incurs substantial training overhead. This is because KD transfers knowledge primarily in the feature space, where the student model is trained from scratch to fit the outputs of a teacher model. Given a student architecture, it is often infeasible to obtain a pre-trained model with the exact same architecture for initialization. The absence of parameter-space knowledge transfer results in slow convergence. In contrast, structured pruning naturally enables parameter-space knowledge transfer by inheriting weights from the pre-trained model, which provides a strong initialization and significantly accelerates convergence compared to KD.

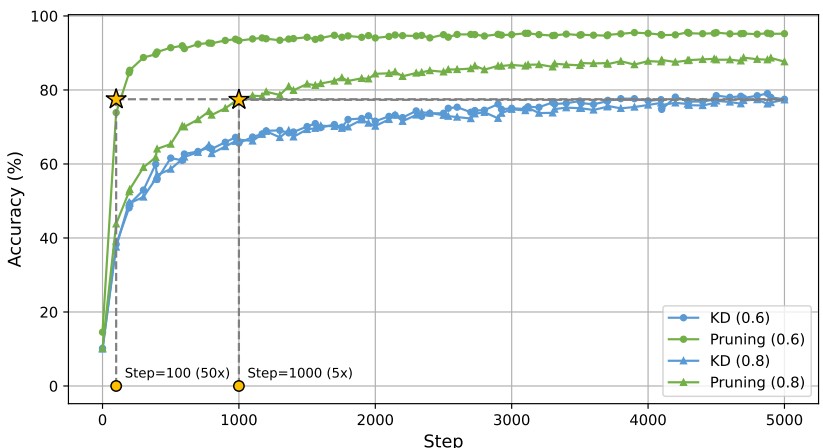

Figure 12: Comparison of convergence speed between logit-based knowledge distillation (KD) and random pruning (Pruning) across different pruning rates with DeiT-Base on CIFAR-10.

To illustrate this, we compare the convergence speed of logit-based KD (Hinton, 2015) with random pruning (Gadhikar et al., 2023) with DeiT-Base on CIFAR-10 (Krizhevsky et al., 2009). Specifically, random pruning uniformly selects a proportion of query-key and value-output dimensions in MHA modules as well as neurons in FFN modules according to the target pruning rate, and removes them to construct a smaller model. In the KD setting, the same student architecture derived from random pruning is instead trained from scratch, with the original pre-trained model serving as the teacher. A KL divergence loss (Van Erven & Harremos, 2014) between the student and teacher logits is used to perform distillation. As shown in Figure 12, the pruned models converge significantly faster than their KD counterparts, highlighting the effectiveness of parameter-space knowledge transfer. Nevertheless, it is worth noting that KD can be effectively combined with structured pruning to further improve the accuracy of pruned models.

## C  INTERPRETABILITY OF NEURONS IN FFN

As discussed in Section 3.2, neurons in the FFN modules of ViTs exhibit remarkably strong interpretability. In this appendix, we provide additional experimental details and supplementary results to further support this finding. We begin by clarifying how the activation of a specific neuron $n_1^{(l,i)} \in \mathbb{R}^d$ is computed for a given image. For an input image, it is represented as a sequence of patch tokens $\mathbf{X} \in \mathbb{R}^{N \times d}$ fed into the FFN of the $l$-th block. The activation of neuron $n_1^{(l,i)}$ for this image is then defined as:

$$a^{(l,i)} = \sum_{j=1}^{N} \sigma\left(\langle \mathbf{X}[j], n_1^{(l,i)} \rangle\right) \tag{17}$$

We use Eq. (17) to measure the relevance between an image and a neuron, and then rank all images in the dataset accordingly. For each neuron, we select the top-25 images with the highest activations to examine the type of knowledge encoded in that neuron.

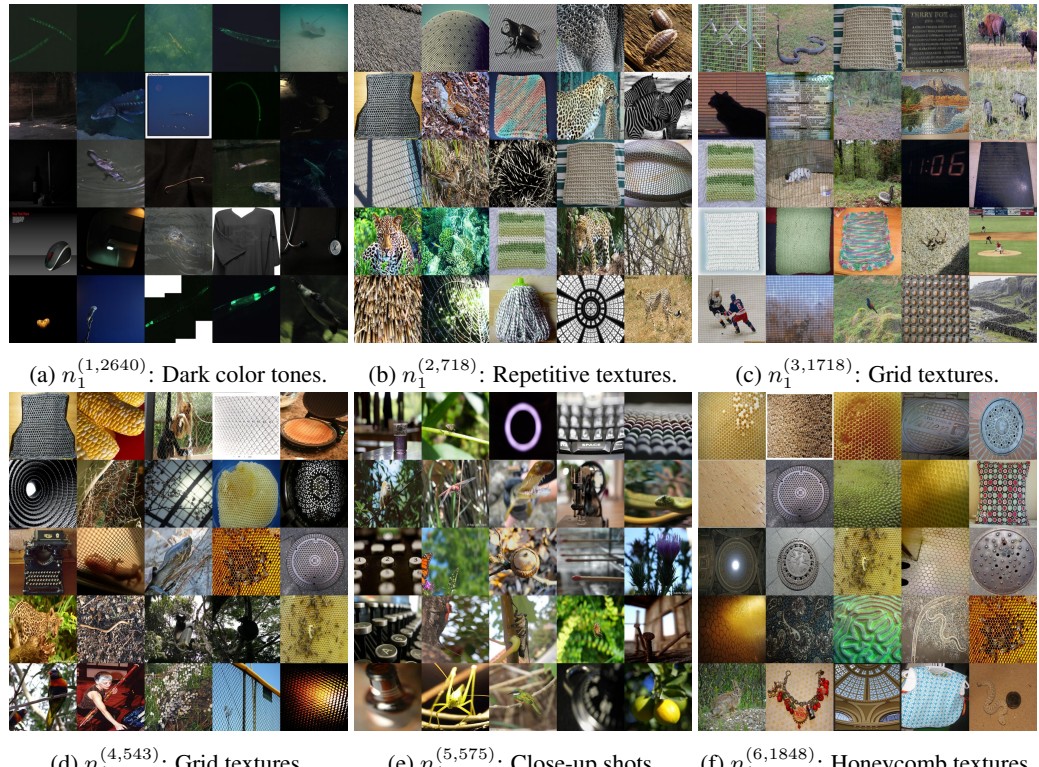

(a) $n_1^{(1,2640)}$: Dark color tones.     (b) $n_1^{(2,718)}$: Repetitive textures.     (c) $n_1^{(3,1718)}$: Grid textures.

(d) $n_1^{(4,543)}$: Grid textures.     (e) $n_1^{(5,575)}$: Close-up shots.     (f) $n_1^{(6,1848)}$: Honeycomb textures.

Figure 13: Additional visualizations of top-25 activated images for randomly selected FFN neurons from the first six blocks of DeiT-Base.

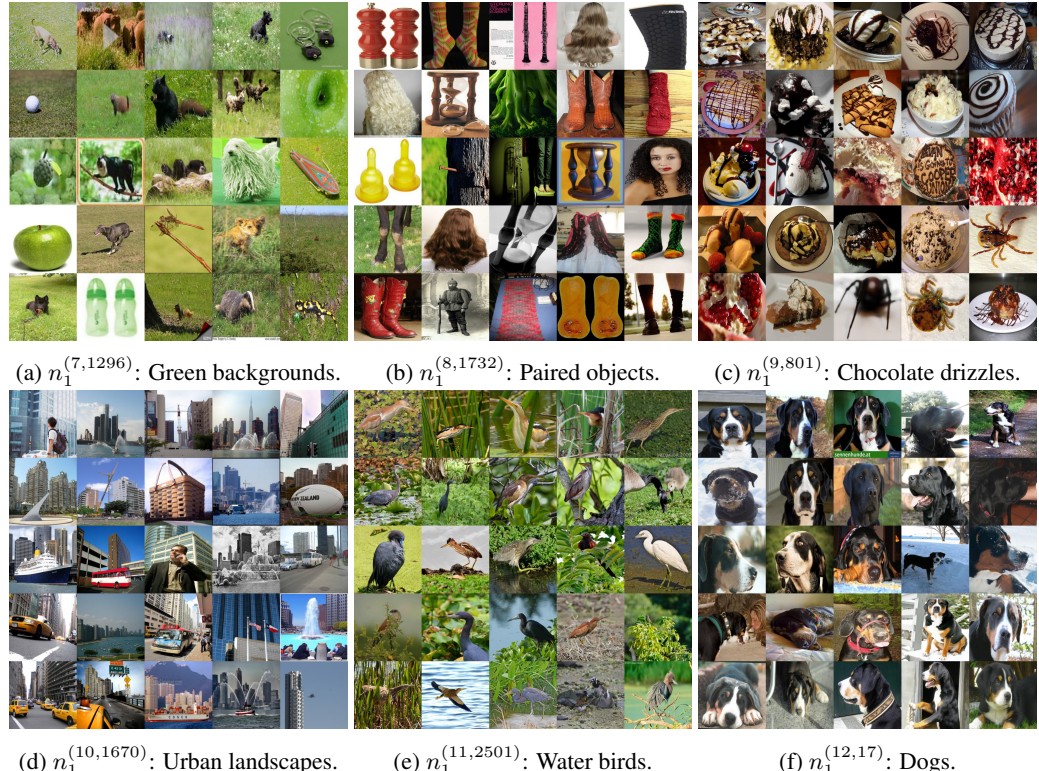

(a) $n_1^{(7,1296)}$: Green backgrounds. (b) $n_1^{(8,1732)}$: Paired objects. (c) $n_1^{(9,801)}$: Chocolate drizzles.

(d) $n_1^{(10,1670)}$: Urban landscapes. (e) $n_1^{(11,2501)}$: Water birds. (f) $n_1^{(12,17)}$: Dogs.

Figure 14: Additional visualizations of top-25 activated images for randomly selected FFN neurons from the last six blocks of DeiT-Base.

As shown in Figures 13 and 14, we provide additional visualizations of randomly selected neurons from all blocks of DeiT-Base. Consistent with the observations in Section 3.2, these neurons exhibit strong interpretability, and the knowledge stored in deeper blocks tends to be more semantic. Interestingly, as illustrated in Figure 13(e), we even observe neurons that capture patterns such as "close-up shots," which are unrelated to the actual content of the image. This further demonstrates that FFN neurons not only encode class-specific knowledge but also capture interpretable patterns beyond classes.

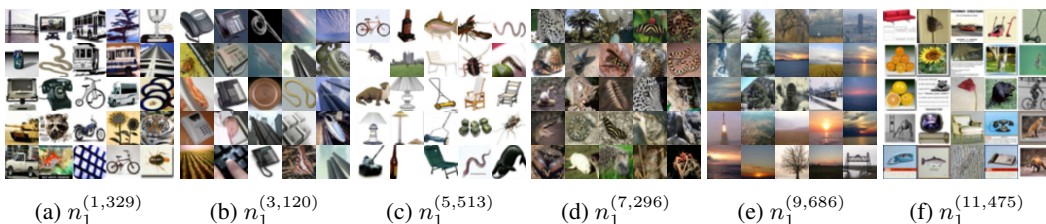

(a) $n_1^{(1,329)}$ (b) $n_1^{(3,120)}$ (c) $n_1^{(5,513)}$ (d) $n_1^{(7,296)}$ (e) $n_1^{(9,686)}$ (f) $n_1^{(11,475)}$

Figure 15: Top-25 activated images (sampled from CIFAR-10) for random FFN neurons in DeiT-Tiny fine-tuned on CIFAR-10.

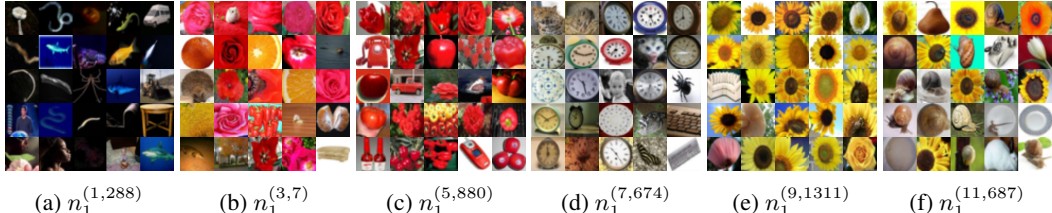

(a) $n_1^{(1,288)}$    (b) $n_1^{(3,7)}$    (c) $n_1^{(5,880)}$    (d) $n_1^{(7,674)}$    (e) $n_1^{(9,1311)}$    (f) $n_1^{(11,687)}$

Figure 16: Top-25 activated images (sampled from CIFAR-100) for random FFN neurons in DeiT-Small fine-tuned on CIFAR-100.

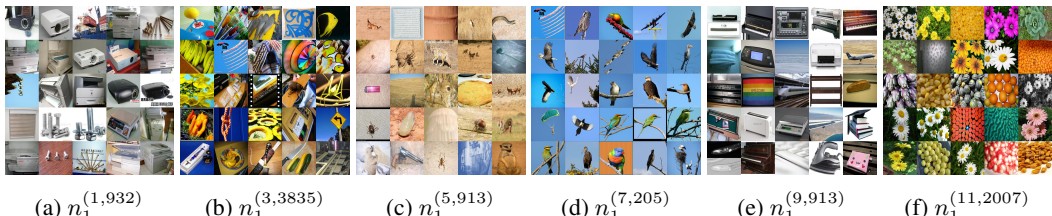

(a) $n_1^{(1,932)}$    (b) $n_1^{(3,3835)}$    (c) $n_1^{(5,913)}$    (d) $n_1^{(7,205)}$    (e) $n_1^{(9,913)}$    (f) $n_1^{(11,2007)}$

Figure 17: Top-25 activated images (sampled from ImageNet-1K) for random FFN neurons in ViT-Large/16.

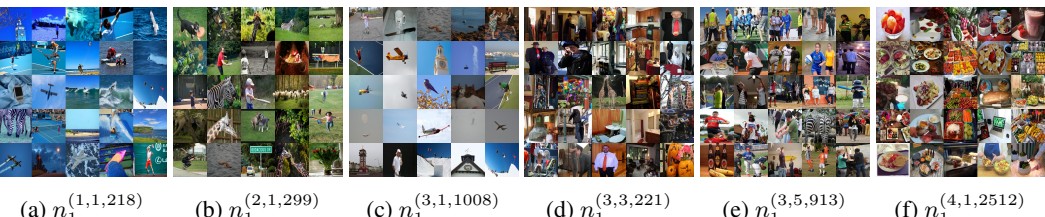

(a) $n_1^{(1,1,218)}$    (b) $n_1^{(2,1,299)}$    (c) $n_1^{(3,1,1008)}$    (d) $n_1^{(3,3,221)}$    (e) $n_1^{(3,5,913)}$    (f) $n_1^{(4,1,2512)}$

Figure 18: Top-25 activated images (sampled from COCO2017) for random FFN neurons in the Swin-Tiny backbone of Mask R-CNN. $n_1^{(s,b,i)}$ denotes the $i$-th neuron in the FFN of the $b$-th block in the $s$-th stage of the Swin Transformer.

To demonstrate that the interpretability of FFN neurons based on their activation patterns is a general phenomenon across ViTs, we conducted additional visualizations on different models and datasets. Specifically, for each model, DeiT-Tiny, DeiT-Small, ViT-Large, and Mask R-CNN (Swin-T), we randomly sampled FFN neurons and visualized the patterns they respond to on CIFAR-10, CIFAR-100, ImageNet, and COCO. As shown in Figures 15-18, these visualizations consistently indicate that the FFN modules of ViT models encode class-specific knowledge.

# D EXTENDING NEURON COLLAPSE TO ALTERNATIVE FFN ARCHITECTURES

Our class-centric neuron collapse (CCNC), originally designed for the canonical FFN architecture, is representative and generalizable, making it applicable to other mainstream Transformer FFN variants. Taking SwiGLU (Shazeer, 2020), a widely used design in LLMs, as an example, its computation is as follows:

$$
\begin{aligned}
\mathrm{SwiGLU}^{(l)}(\mathbf{X}) &= \left(\sigma(\mathbf{X}W_1^{(l)\top}) \odot (\mathbf{X}W_g^{(l)\top})\right) W_2^{(l)\top} \\
&= \sum_{i=1}^{e_l} \left(\sigma(\mathbf{X}W_1^{(l)}[i]) \odot (\mathbf{X}W_g^{(l)}[i])\right) \otimes W_2^{(l)\top}[i] \\
&= \sum_{i=1}^{e_l} \left(\sigma(\mathbf{X}n_1^{(l,i)}) \odot (\mathbf{X}n_g^{(l,i)})\right) \otimes n_2^{(l,i)}
\end{aligned}
\tag{18}
$$

where $W_1^{(l)} \in \mathbb{R}^{e_l \times d}$ and $W_2^{(l)} \in \mathbb{R}^{d \times e_l}$ are the up- and down-projection matrices of the FFN in the $l$-th block of ViT, $W_g \in \mathbb{R}^{e_l \times d}$ is the gating matrix, and $n_{1|2|g}^{(l,i)} \in \mathbb{R}^d$ denotes the $i$-th neuron in $W_{1|2|g}^{(l)}$. To prune the intermediate dimension $e_l$ of the FFN, one only needs to extend Eq. (4) with additional constraints on $n_g$. Specifically, the anchor neuron of each cluster for $n_1$ and $n_g$ is selected as the one that maximizes the following value:

$$s^{(l,i)} = \sum_{j=1}^{N} \left( \sigma \left( \langle \mathbf{X}[j], n_1^{(l,i)} \rangle \right) \odot \langle \mathbf{X}[j], n_g^{(l,i)} \rangle \right) \tag{19}$$

Notably, in most Transformer architectures, the neurons associated with different intermediate dimensions of the FFN (e.g., $n_1^{(l,i)}$, $n_2^{(l,i)}$, and $n_g^{(l,i)}$ at the $i$-th dimension in SwiGLU) are mutually independent, which implies that CCNC can be readily applied to these architectures as well.

## E COMPARISON OF PRUNING SETTINGS IN MHA

Table 6: Comparison of head-level and dimension-level pruning on DeiT-Base with ImageNet-1K sub-tasks under pruning rates of 0.40 and 0.60. Dimension pruning achieves consistently higher accuracy, showing its advantage in preserving knowledge compared to coarse-grained head pruning.

| Methods | T1/25 | | T2/50 | | T3/100 | |
|---|---|---|---|---|---|---|
| | 0.40 | 0.60 | 0.40 | 0.60 | 0.40 | 0.60 |
| Head-level | 4.56 | 1.12 | 1.68 | 0.92 | 14.11 | 3.47 |
| Dimension-level | 78.08 | 59.76 | 78.76 | 62.20 | 77.29 | 61.46 |

**Head or Dimension?** The Multi-Head Attention (MHA) module can be pruned at two different granularities: head-level pruning, which removes entire attention heads, and dimension-level pruning, which reduces the dimensionality of the query, key, or value vectors. To determine the appropriate pruning granularity for Vulcan, we conduct a comparative study between these two strategies. For head-level pruning, we follow prior works and adopt the Taylor expansion approximation (Molchanov et al., 2019; Yang et al., 2023) criterion to evaluate the importance of each attention head and perform block-uniform pruning (Blalock et al., 2020). For dimension-level pruning, we apply singular value decomposition (SVD) to compress the query-key (QK) and value-output (VO) dimensions, motivated by the clear advantage of SVD observed in Figure 4.

As shown in Table 6, we compare these two strategies across different pruning rates (0.40, 0.60) and sub-task (25, 50, 100) sizes with DeiT-Base on ImageNet-1K. The results show that dimension-level pruning consistently outperforms head-level pruning, indicating that coarse-grained head pruning leads to substantial knowledge loss. In contrast, Vulcan leverages the computational structure of MHA and applies SVD-based adaptive pruning to the QK and VO dimensions, enabling more efficient and effective knowledge extraction.

Table 7: Inference latency of DeiT-Base, DeiT-Small, and DeiT-Tiny under uniform vs. uneven QK/VO dimensions on Jetson Orin NX (bz=1) and NVIDIA A40 (bz=256). In the uneven setting, the dimensions of each head in each block are randomly assigned while keeping the total parameter count unchanged.

| Methods | DeiT-Base (ms) | | DeiT-Small (ms) | | DeiT-Tiny (ms) | |
|---|---|---|---|---|---|---|
| | bz=1 | bz=256 | bz=1 | bz=256 | bz=1 | bz=256 |
| Uniform | 45.45 | 600.52 | 15.58 | 186.30 | 14.52 | 76.45 |
| Uneven | 84.10 | 629.62 | 50.66 | 200.63 | 31.50 | 82.51 |

**Uniform *vs*. Uneven QK/VO Dimensions**. One key design of Vulcan is to enforce identical query-key (QK) and value-output (VO) dimensions across all heads within the same block, which is crucial for efficient inference. When head dimensions are uniform, the MHA module can be computed in parallel across heads; otherwise, attention must be evaluated sequentially, introducing slicing operations and memory copies that significantly increase latency. As shown in Table 7, we measure

inference latency on both Jetson Orin NX (batch size = 1) and NVIDIA A40 (batch size = 256) for DeiT-Base, DeiT-Small, and DeiT-Tiny under two settings: uniform vs. uneven QK/VO dimensions. Despite identical parameter counts and GFLOPs, the uniform-head setting achieves consistently lower latency, with the gap particularly pronounced on edge devices.

## F  THEORETICAL ANALYSIS OF SVD FOR MHA COMPRESSION

As shown in Figure 4, pruning the query-key and value-output dimensions with singular value decomposition (SVD) achieves significantly better performance than score-based pruning methods. In this section, we first introduce the fundamentals of SVD, and then provide a theoretical explanation for this seemingly counterintuitive observation.

**SVD**. Given a matrix $W \in \mathbb{R}^{m \times n}$, SVD factorizes it into a set of orthogonal singular vectors and their associated singular values, which capture the intrinsic low-rank structure of the matrix. Formally, SVD expresses $W$ as:

$$W = U\Sigma V^\top, \quad U \in \mathbb{R}^{m \times m}, \Sigma \in \mathbb{R}^{m \times n}, V \in \mathbb{R}^{n \times n} \tag{20}$$

where $U$ is the left singular matrix, $\Sigma$ is the diagonal singular value matrix, and $V$ is the right singular matrix. The columns of $U$, referred to as the left singular vectors, are eigenvectors of $WW^\top$ and provide an orthogonal basis for the input (row) space. The columns of $V$, referred to as the right singular vectors, are eigenvectors of $W^\top W$ and provide an orthogonal basis for the output (column) space. The diagonal entries of $\Sigma$ are non-negative real numbers known as singular values, arranged in descending order. Each singular value quantifies the "stretching factor" of the matrix along the corresponding singular direction, with larger values indicating more informative directions.

**Why SVD is Better**?  SVD not only reveals the low-rank structure of $W$ but also allows for constructing its best rank-$k$ approximation by truncating the top-$k$ singular values and their associated singular vectors. Taking $W_Q \in \mathbb{R}^{q \times d}$ and $W_K \in \mathbb{R}^{q \times d}$ as an example, when pruning the query-key dimension, the goal is to preserve the intermediate representations as much as possible. In other words, the optimal pruned matrices $W_Q'^* \in \mathbb{R}^{q' \times d}$ and $W_K'^* \in \mathbb{R}^{q' \times d}$ should satisfy:

$$W_Q'^*, W_K'^* = \underset{W_Q', W_K'}{\arg\min} \|W_Q'^\top W_K' - W_Q^\top W_K\|_F \tag{21}$$

According to the *Eckart-Young theorem* (Chipman, 2020), SVD guarantees the minimization of the truncation error in the Frobenius norm. Specifically, for a given pruned query-key dimension $q' < q$, the optimal pruned matrices $W_Q'$ and $W_K'$ are obtained by applying SVD to $W_Q^\top W_K$:

$$W_Q^\top W_K = U_{QK} \Sigma_{QK} V_{QK}^\top \tag{22}$$

$$W_Q'^* = (U_{QK}[:,:q']\Sigma_{QK}[:q',:q'])^\top, \quad W_K'^* = (V_{QK}[:,:q'])^\top \tag{23}$$

In contrast, score-based pruning methods remove certain row vectors from $W_Q$ and $W_K$ directly, which leads to a truncated error in the Frobenius norm that is no smaller than that obtained by SVD. This inevitably results in greater loss of information. Therefore, SVD maximizes knowledge preservation under the same pruning rate, which explains its significant advantage over score-based methods.

## G  PROOF OF PRUNING RATE SATISFACTION IN DERIVED MODELS

The pruning rate $R$ is determined according to the resource constraints of the target edge device. To ensure that the derived models can be successfully deployed for inference on such devices, Vulcan must strictly satisfy the specified pruning rate. In this section, we first introduce the computation of FLOPs for ViT, and then provide a formal proof that the size of the models derived by Vulcan exactly matches the pruning rate $R$.

**FLOPs**. FLOPs represent the number of floating-point operations required for a model to perform inference on a single image. For a ViT, let the embedding dimension be $d$, the query–key dimension, value–output dimension, and the number of heads in the MHA module of the $l$-th block be $q_l$, $v_l$, and $H_l$, respectively, and let the intermediate dimension of the FFN module in the $l$-th block be $e_l$.

Then, for a ViT $\mathcal{M}_B$ consisting of $L$ blocks, the total FLOPs can be formulated as:

$$\text{FLOPs}(\mathcal{M}_B) = \sum_{l=1}^{L} \left[ \left(2Ndq_l + 2Ndv_l + N^2 q_l + N^2 v_l\right) \times H_l + (2Nde_l) \right]$$

$$= (2Nd + N^2) \sum_{l=1}^{L} H_l(q_l + v_l) + 2Nd \sum_{l=1}^{L} e_l \tag{24}$$

where $N$ is the number of patch tokens. It can be observed that the three dimensions pruned by Vulcan, i.e., $q_l$, $v_l$, and $e_l$, are directly associated with the FLOPs of a ViT.

***Proof2: Pruning Rate Satisfaction.*** We denote the actual pruning rate of the derived model $\mathcal{M}_E$ as $R'$. Then, proving that Vulcan strictly satisfies the target pruning rate amounts to showing that:

$$R' = 1 - \frac{\text{FLOPs}(\mathcal{M}_E)}{\text{FLOPs}(\mathcal{M}_B)} \geq R \tag{25}$$

From Eq. (3), it follows that the intermediate dimension of the pruned FFN module in the $l$-th block, denoted $e_l'$, satisfies:

$$\sum_{l=1}^{L} e_l' = \sum_{l=1}^{L} K^{(l)} = \sum_{l=1}^{L} \sum_{i=1}^{e_l} \mathbb{I}\left( a^{(l,i)} > \Phi\left(\mathcal{A}, \lceil (\sum_{j=1}^{L} e_j) \times R \rceil\right)\right)$$

$$= \sum_{l=1}^{L} e_l - \lceil (\sum_{l=1}^{L} e_l) \times R \rceil < \sum_{l=1}^{L} e_l - (\sum_{l=1}^{L} e_l) \times R \tag{26}$$

$$= \left(\sum_{l=1}^{L} e_l\right) \times (1 - R)$$

Under the assumption that $q_l = v_l$ and $q_i = q_j$, according to Eq. (5) and Eq. (6), the query–key dimension $q_l'$ and value–output dimension $v_l'$ in the pruned MHA modules satisfy:

$$\sum_{l=1}^{L}(q_l' + v_l') = \sum_{l=1}^{L}\left( \lfloor q_l(1 - R_{QK}^{(l)})\rfloor + \lfloor v_l(1 - R_{VO}^{(l)})\rfloor\right)$$

$$< \sum_{l=1}^{L}\left( q_l(1 - R_{QK}^{(l)}) + v_l(1 - R_{VO}^{(l)})\right)$$

$$= \sum_{l=1}^{L}\left( \frac{r_{QK}^{(l)} + r_{VO}^{(l)} - 2r_{VO}^{(l)}R^{(l)}}{r_{QK}^{(l)} + r_{VO}^{(l)}} q_l + \frac{r_{QK}^{(l)} + r_{VO}^{(l)} - 2r_{QK}^{(l)}R^{(l)}}{r_{QK}^{(l)} + r_{VO}^{(l)}} v_l \right)$$

$$= \sum_{l=1}^{L} \frac{2(r_{QK}^{(l)} + r_{VO}^{(l)}) - 2(r_{QK}^{(l)} + r_{VO}^{(l)})R^{(l)}}{r_{QK}^{(l)} + r_{VO}^{(l)}} q_l$$

$$= 2\sum_{l=1}^{L}(1 - R^{(l)})q_l = 2q_1 \sum_{l=1}^{L}(1 - R^{(l)}) \tag{27}$$

$$= 2q_1 \sum_{l=1}^{L} \left( 1 - \frac{L}{\sum_{i=1}^{L} 1/(r_{QK}^{(i)} + r_{VO}^{(i)})} \cdot \frac{R}{r_{QK}^{(l)} + r_{VO}^{(l)}} \right)$$

$$= 2Lq_1 - 2q_1 \frac{LR \sum_{l=1}^{L} 1/(r_{QK}^{(l)} + r_{VO}^{(l)})}{\sum_{i=1}^{L} 1/(r_{QK}^{(i)} + r_{VO}^{(i)})} = 2Lq_1 - 2Lq_1 R$$

$$= 2Lq_1 \cdot (1 - R) = (L(q_1 + v_1))(1 - R) = \left( \sum_{l=1}^{L}(q_l + v_l)\right)(1 - R)$$

Under the assumption that $H_i = H_j$, substituting these relations into Eq. (25), we obtain:

$$
\begin{aligned}
R' &= 1 - \frac{(2Nd + N^2)\sum_{l=1}^{L} H_l(q_l' + v_l') + 2Nd\sum_{l=1}^{L} e_l'}{(2Nd + N^2)\sum_{l=1}^{L} H_l(q_l + v_l) + 2Nd\sum_{l=1}^{L} e_l} \\
&< 1 - \frac{(2Nd + N^2)\sum_{l=1}^{L} H_l(q_l + v_l)(1 - R) + 2Nd\sum_{l=1}^{L} e_l(1 - R)}{(2Nd + N^2)\sum_{l=1}^{L} H_l(q_l + v_l) + 2Nd\sum_{l=1}^{L} e_l} \\
&= 1 - (1 - R) = R
\end{aligned}
\tag{28}
$$

This establishes that the models derived by Vulcan strictly satisfy the specified pruning rate $R$. $\qquad\square$

## H    IMPLEMENTATION DETAILS

We summarize the hyperparameter settings used during the post-training process in Table 8.

Table 8: Hyperparameters for the post-training process of Vulcan. Here, $R$ denotes the target pruning rate and $\lambda$ represents the Lagrange multipliers.

| Hyperparam | Value |
|---|---|
| Steps | $6250R^2 + 1250R, \quad R \in (0,1)$ |
| Optimizer | AdamW (Loshchilov & Hutter, 2017) |
| Batch Size | 256 (Recognition), 8 (Detection/Segmentation) |
| Learning Rate (LR) | 1e-4 (Recognition), 5e-5 (Detection/Segmentation) |
| LR Scheuler | Constant |
| Weight Decay | 0.05 |
| Penalty Parameter $\rho$ | 1.0 |
| Penalty Update Rule | $\lambda_{i+1} = \lambda_i + \rho(\partial\mathcal{L}/\partial\lambda_i)$ |
| Seed | 42 |

**Normalization**. As discussed in Section 4, Vulcan will normalize the activations of each neuron before computing $K^{(l)}$. The choice of normalization strategy is closely related to the derived model architecture. Therefore, we compare three commonly used normalization strategies. For activation vectors $a^{(l)} = [a^{(l,1)}, a^{(l,2)}, \ldots, a^{(l,e_l)}] \in \mathbb{R}^{e_l}$, their normalization processes are as follows:

- L2 Normalization:

$$
a_{\text{norm}}^{(l,i)} = \frac{a^{(l,i)}}{\sqrt{\sum_{i=1}^{e_l}\left(a^{(l,i)}\right)^2}}
$$

- Min-Max Normalization:

$$
a_{\text{norm}}^{(l,i)} = \frac{a^{(l,i)} - \min(a^{(l)})}{\max(a^{(l)}) - \min(a^{(l)})}
$$

- Z-score Normalization:

$$
a_{\text{norm}}^{(l,i)} = \frac{a^{(l,i)} - \mu}{\sigma}, \quad \mu = \frac{1}{e_l}\sum_{j=1}^{e_l} a^{(l,j)}, \quad \sigma = \sqrt{\sum_{j=1}^{e_l}\left(a^{(l,j)} - \mu\right)^2}
$$

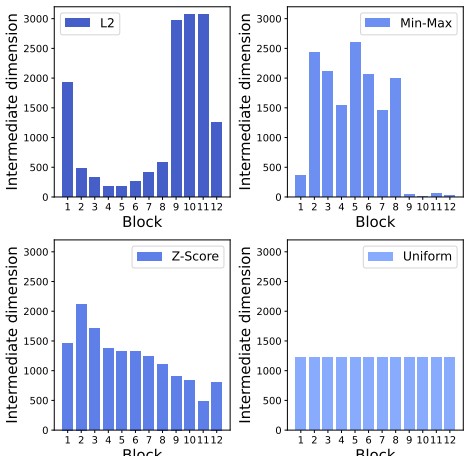 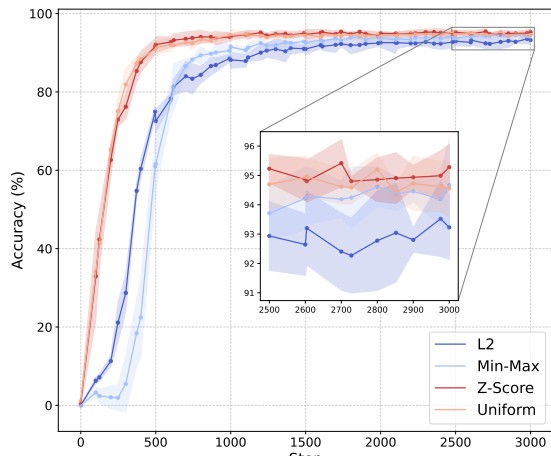

Figure 19: Architectures of models derived under different normalization strategies. Since the architectures corresponding to T1–T3 are highly similar, this figure shows their averaged architectures.

Figure 20: Convergence curves of models with three normalization methods and the uniform architecture. Z-score normalization and the uniform architecture achieve faster convergence and higher accuracy than L2 and Min-Max normalization.

At a pruning rate of 0.60, we derive a series of edge ViTs from DeiT-Base for the three ImageNet sub-tasks (T1–T3) in Table 1. In addition, we introduce uniformly structured models as a baseline for comparison with the activation-adaptive architectures. As illustrated in Figure 19, the architectures resulting from different normalization methods exhibit substantial differences. Specifically, L2 normalization produces architectures with fewer parameters in the intermediate blocks, min-max normalization leads to architectures with fewer parameters in the deeper blocks, while z-score normalization yields relatively uniform architectures, showing an overall trend where the number of parameters decreases as the block depth increases. Figure 20 further demonstrates that the architecture derived with z-score normalization achieves the best performance, and thus Vulcan adopts it for normalizing activation values of DeiT. At the same time, we also observe that the effectiveness of normalization strategies is model-dependent. For Swin-Transformer, Vulcan instead adopts min-max normalization.

**Adaptive Pruning Rate Allocation for Non-Uniform Architectures**. Eq. (3) and Eq. (5) implicitly assume that all Transformer blocks share an identical architecture, which does not hold for non-uniform models such as Swin-Transformer (Liu et al., 2021), where different blocks may contain heterogeneous structures. To address this limitation, we extend Eq. (3) and Eq. (5) to support non-uniform architectures. First, let the number of parameters in the FFN module and the MHA module of the $l$-th block be denoted as $P_F^{(l)}$ and $P_M^{(l)}$, respectively. By incorporating $P_F^{(l)}$ and $P_M^{(l)}$ into the importance evaluation, we generalize the computation of $K^{(l)}$ and $R^{(l)}$ as follows:

$$K^{(l)} = \sum_{i=1}^{e_l} \mathbb{I}\left( \frac{a^{(l,i)}}{\log(P_F^{(l)})} > \Phi\big(\mathcal{A}, \lceil (\sum_{j=1}^{L} e_j) \times R \rceil\big) \right), \quad \mathcal{A} = \bigcup_{l=1}^{L} \{ \frac{a^{(l,i)}}{\log(P_F^{(l)})} | i = 1, \ldots, e_l \} \quad (29)$$

$$R^{(l)} = \frac{\sum_{i=1}^{L} P_M^{(i)}}{\sum_{i=1}^{L} P_M^{(i)} \left( P_M^{(i)} / (r_{QK}^{(i)} + r_{VO}^{(i)}) \right)} \cdot \frac{P_M^{(l)}}{r_{QK}^{(l)} + r_{VO}^{(l)}} \cdot R \quad (30)$$

It is worth noting that in certain extreme cases (e.g., when the importance of a particular block is significantly lower than that of others while the pruning rate is very high), the values of $R^{(l)}$ computed by Eq. (5) and Eq. (30) may exceed 1. In such situations, Vulcan sets the query–key and value–output dimensions of the overflowing blocks to 1, and iteratively recalculates the $R^{(l)}$ of the remaining blocks until all blocks satisfy $R^{(l)} \in (0, 1)$.

## I    BASELINES

In this section, we detail the five baselines used for comparison with Vulcan:

- **Random**. The random pruning method operates on the same dimensions as Vulcan, i.e., the intermediate dimension ($e$) of the FFN and the query–key and value–output dimensions ($q, v$) of the MHA. Given a pruning rate, it uniformly prunes dimensions by randomly selecting neurons to delete, without considering the specific sub-task. We adopt this class-agnostic pruning method as a lower-bound benchmark to evaluate the effectiveness of class-specific model derivation.

- **NViT** (Yang et al., 2023). NViT prunes all dimensions of a ViT, including the intermediate dimension ($e$) of the FFN, the query–key, value–output dimensions ($q, v$), and the number of attention heads ($H$) of the MHA, and the embedding dimension ($d$). It groups the weights along each dimension and evaluates their importance using a Hessian-based importance score. The pruning process is performed iteratively by removing the groups with the lowest scores until the resource constraints of the target device are satisfied.

- **X-Pruner** (Yu & Xiang, 2023). X-Pruner prunes the intermediate dimension of the FFN ($e$) and the number of attention heads ($H$) in the MHA. It introduces learnable masks into the ViT, and pruning is achieved by applying sparsity regularization to gradually drive parts of the mask values to zero.

- **DC-ViT** (Zhang et al., 2024). The goal of DC-ViT is to achieve the target pruning rate while pruning as few blocks as possible. It designs an importance metric to evaluate each block, and for the selected blocks, it removes the entire MHA module and uniformly prunes the intermediate dimension ($e$) of the FFN at random. Among all the baselines, DC-ViT is the only method that performs pruning at the module level.

- **MDP** (Sun et al., 2025). Similar to NViT, MDP jointly prunes multiple dimensions of ViTs, including the embedding dimension ($d$), the number of attention heads ($H$), the query–key and value–output dimensions ($q, v$), and the FFN intermediate dimension ($e$). It formulates pruning as a mixed-integer nonlinear program (MINLP) problem under latency budgets, solved with Hessian-based importance scores and a precomputed latency lookup table (LUT).

It is worth noting that although Random pruning is intuitively expected to perform the worst, it actually outperforms some score-based methods. We attribute this to the fact that it adopts the same pruning dimensions and granularity as Vulcan. In contrast, DC-ViT suffers from overly coarse pruning granularity, making it difficult to recover model accuracy through retraining when the pruning rate becomes high.

## J    ADDITIONAL EXPERIMENTAL RESULTS

In this section, we present additional experimental results and analyses. Appendix J.1 compares Vulcan with five baselines at lower pruning rates to further demonstrate the effectiveness of Vulcan. Appendix J.2 visualizes the model architectures of edge ViTs derived by Vulcan and further investigates how class-specific knowledge is distributed across different blocks. Appendix J.3 compares the computational efficiency of Vulcan-derived models with five baselines, demonstrating the edge-friendliness of Vulcan. Appendix J.4 further demonstrates the scalability of Vulcan on larger ViTs and BERT. Appendix J.5 shows that Vulcan enhances model generalization and robustness. Appendix J.6 evaluates Vulcan in complex open-domain scenarios. Appendix J.8 investigates the trade-off between derivation performance and derivation overhead. Appendix J.9 illustrates how Vulcan can be combined with other compression techniques to achieve extreme lightweighting.

## J.1 PERFORMANCE UNDER MODERATE PRUNING

Table 9: Overall performance of Vulcan and baselines on DeiT-Base with ImageNet-1K sub-tasks of different sizes (25, 50, 100 classes) under pruning rates of 0.20 and 0.40.

| Method | T1/25 | T2/25 | T3/25 | Avg | T4/50 | T5/50 | T6/50 | Avg | T7/100 | T8/100 | T9/100 | Avg |
|---|---|---|---|---|---|---|---|---|---|---|---|---|
| **DeiT-Base** | 79.92 | 82.56 | 80.67 | 81.05 | 80.48 | 79.64 | 84.96 | 81.69 | 80.49 | 84.12 | 78.58 | 81.06 |
| **DeiT-Base (FT)** | 96.40 | 96.96 | 96.96 | 96.77 | 95.00 | 93.36 | 94.68 | 94.35 | 91.56 | 93.29 | 92.84 | 92.56 |
| *Pruning Rate = 0.20* | | | | | | | | | | | | |
| **Random** | 95.60 | 96.48 | 96.40 | 96.16 | 93.84 | 93.36 | 94.60 | 93.93 | 91.41 | 93.07 | 91.44 | 91.97 |
| **NViT** | 94.96 | 96.32 | 96.01 | 95.76 | 92.96 | 92.72 | 94.32 | 93.10 | 91.27 | 92.83 | 91.84 | 91.98 |
| **X-Pruner** | 96.56 | 96.48 | 96.88 | 96.62 | 94.40 | 93.00 | 95.40 | 94.26 | 91.16 | 92.65 | 91.70 | 91.84 |
| **DC-ViT** | 95.52 | 95.76 | 94.57 | 95.28 | 91.40 | 93.16 | 94.00 | 92.85 | 84.73 | 88.34 | 90.84 | 87.97 |
| **MDP** | 95.44 | 96.56 | 94.73 | 95.58 | 92.60 | 88.00 | 92.48 | 91.03 | 84.50 | 92.79 | 90.74 | 89.34 |
| *Vulcan* | 96.48 | 97.12 | 97.04 | **96.88** | 95.00 | 93.24 | 94.60 | **94.28** | 91.49 | 92.91 | 91.74 | **92.05** |
| *Pruning Rate = 0.40* | | | | | | | | | | | | |
| **Random** | 94.64 | 95.68 | 94.48 | 94.93 | 91.84 | 90.84 | 92.48 | 91.72 | 87.94 | 90.41 | 89.12 | 89.16 |
| **NViT** | 92.08 | 94.24 | 94.17 | 93.50 | 90.60 | 90.00 | 91.52 | 90.71 | 87.23 | 90.10 | 87.88 | 88.40 |
| **X-Pruner** | 94.76 | 94.68 | 95.01 | 94.82 | 92.34 | 91.60 | 93.28 | 92.41 | 88.95 | 91.67 | 89.04 | 89.89 |
| **DC-ViT** | 86.80 | 89.12 | 88.66 | 88.19 | 83.84 | 80.60 | 87.52 | 83.99 | 80.91 | 83.73 | 83.16 | 82.60 |
| **MDP** | 95.36 | 96.40 | 93.69 | 95.15 | 93.40 | 89.76 | 90.60 | 91.25 | 88.89 | 88.12 | 89.36 | 88.79 |
| *Vulcan* | 94.80 | 95.68 | 94.97 | **95.15** | 92.76 | 91.52 | 93.60 | **92.63** | 89.71 | 91.59 | 89.84 | **90.38** |

We first compare Vulcan with five baselines at pruning rates of 0.20 and 0.40 to evaluate its performance under moderate pruning conditions. As shown in Table 9, we observe that at pruning rates of 0.20 and 0.40, Vulcan consistently outperforms the base ViT and five baselines in terms of accuracy across different sub-task sizes. Specifically, at a pruning rate of 0.20, the model derived by Vulcan outperforms the base ViT by 13.13% on average, and further surpass the best-performing baselines, X-Pruner and MDP, by up to 0.64% and 6.99%, respectively. At the higher pruning rate of 0.40, Vulcan achieves average accuracy improvements of 11.45%, 0.35%, and 5.99% over the base ViT, X-Pruner, and MDP, respectively. These results further confirm the effectiveness of Vulcan, showing that it delivers consistent advantages even under moderate pruning conditions.

## J.2 MODEL ARCHITECTURES OF EDGE VITS

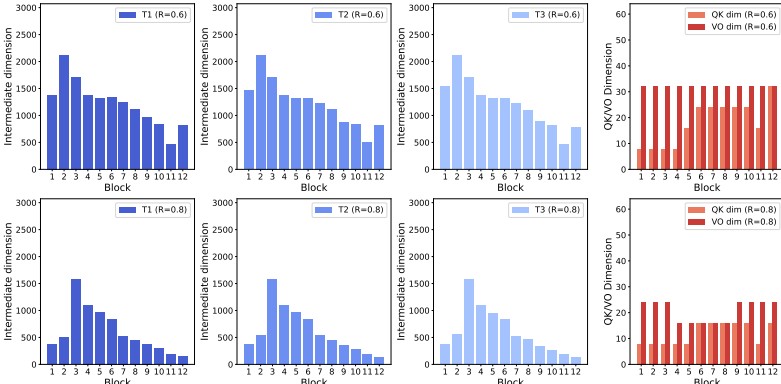

Figure 21: Visualization of the architectures of edge ViTs derived by Vulcan from DeiT-Base for sub-tasks T1–T3 on ImageNet under pruning rates of 0.60 and 0.80.

We visualize the architectures of edge ViTs derived from DeiT-Base for the sub-tasks T1–T3 in Table 1, as shown in Figure 21. From the visualization, we observe that class-specific knowledge in the FFN modules is more concentrated in the middle blocks, while class-agnostic knowledge in the MHA modules is more concentrated in the value–output dimensions.

## J.3 COMPUTATIONAL EFFICIENCY COMPARISON

Table 10: Comparison of computational efficiency between DeiT-Base and edge ViTs derived by Vulcan and five baselines under different pruning rate $R$, evaluated on Jetson Orin NX (bz=1) and NVIDIA RTX 4090 (bz=256).

| Methods | Latency (ms) | | Throughput (image/s) | | Memory (GB) | | #Param (M) | #FLOPs (G) | Acc. (%) |
|---|---|---|---|---|---|---|---|---|---|
| | Orin NX | RTX 4090 | Orin NX | RTX 4090 | Orin NX | RTX 4090 | | | |
| **DeiT-Base** | 45.45 | 274.27 | 22.00 | 933.39 | 0.34 | 2.21 | 86.57 | 17.57 | 81.27 |
| *Pruning Rate = 0.20* | | | | | | | | | |
| **Random** | 42.97 (1.05×) | 278.21 (0.99×) | 23.27 | 920.15 | 0.28 | 2.75 | 69.47 | 14.06 (↓19.98%) | 94.02 (↑12.75) |
| **NViT** | 37.49 (1.21×) | 227.53 (1.21×) | 26.68 | 1135.15 | 0.27 | 2.15 | 68.71 | 14.05 (↓19.98%) | 93.61 (↑12.34) |
| **X-Pruner** | 37.49 (1.21×) | 223.87 (1.23×) | 26.67 | 1143.51 | 0.28 | 2.15 | 68.58 | 13.94 (↓20.66%) | 94.24 (↑12.97) |
| **DC-ViT** | 36.98 (1.23×) | 225.19 (1.22×) | 27.04 | 1136.83 | 0.28 | 2.15 | 69.62 | 12.49 (↓28.91%) | 92.03 (↑10.76) |
| **MDP** | 33.78 (1.35×) | 202.48 (1.35×) | 29.61 | 1264.31 | 0.25 | 2.12 | 62.21 | 14.05 (↓19.98%) | 91.98 (↑10.71) |
| **Vulcan** | 36.84 (1.23×) | 218.16 (1.26×) | 27.14 | 1173.43 | 0.27 | 2.14 | 67.22 | 13.56 (↓21.51%) | 94.40 (↑13.13) |
| *Pruning Rate = 0.40* | | | | | | | | | |
| **Random** | 34.24 (1.33×) | 229.04 (1.20×) | 29.20 | 1117.69 | 0.21 | 2.57 | 52.39 | 10.55 (↓39.95%) | 91.94 (↑10.67) |
| **NViT** | 29.25 (1.55×) | 182.20 (1.51×) | 34.19 | 1405.02 | 0.21 | 1.99 | 50.86 | 10.54 (↓40.01%) | 90.87 (↑9.60) |
| **X-Pruner** | 29.54 (1.54×) | 183.33 (1.49×) | 33.86 | 1396.38 | 0.21 | 2.09 | 52.61 | 10.71 (↓39.04%) | 92.37 (↑11.10) |
| **DC-ViT** | 28.62 (1.59×) | 175.33 (1.56×) | 34.94 | 1460.09 | 0.21 | 2.08 | 52.38 | 10.54 (↓39.04%) | 84.93 (↑3.66) |
| **MDP** | 26.30 (1.73×) | 159.68 (1.72×) | 38.03 | 1603.25 | 0.19 | 2.06 | 45.76 | 9.25 (↓47.35%) | 91.73 (↑10.46) |
| **Vulcan** | 29.73 (1.53×) | 176.01 (1.56×) | 33.63 | 1454.42 | 0.21 | 1.82 | 51.00 | 10.24 (↓41.72%) | 92.72 (↑11.45) |
| *Pruning Rate = 0.60* | | | | | | | | | |
| **Random** | 26.74 (1.70×) | 184.40 (1.49×) | 37.39 | 1388.29 | 0.15 | 2.39 | 35.29 | 7.04 (↓59.93%) | 87.01 (↑5.74) |
| **NViT** | 24.55 (1.85×) | 137.03 (2.00×) | 40.73 | 1868.23 | 0.14 | 1.44 | 33.13 | 7.02 (↓60.05%) | 81.88 (↑0.61) |
| **X-Pruner** | 21.79 (2.09×) | 135.53 (2.02×) | 45.89 | 1888.93 | 0.15 | 2.02 | 35.56 | 7.27 (↓58.62%) | 89.41 (↑8.14) |
| **DC-ViT** | 20.04 (2.27×) | 126.84 (2.16×) | 49.90 | 2018.22 | 0.15 | 2.02 | 35.43 | 7.03 (↓59.99%) | 69.85 (↓11.42) |
| **MDP** | 19.35 (2.35×) | 122.99 (2.23×) | 51.67 | 2081.54 | 0.13 | 1.43 | 31.08 | 6.20 (↓64.71%) | 88.25 (↑6.98) |
| **Vulcan** | 21.81 (2.16×) | 136.67 (2.01×) | 45.86 | 1873.11 | 0.15 | 1.66 | 34.09 | 6.77 (↓61.47%) | 92.32 (↑11.05) |
| *Pruning Rate = 0.80* | | | | | | | | | |
| **Random** | 15.19 (2.99×) | 101.99 (2.69×) | 65.85 | 2510.1 | 0.08 | 1.06 | 18.21 | 3.53 (↓79.91%) | 71.45 (↓9.82) |
| **NViT** | 19.13 (2.37×) | 98.94 (2.77×) | 52.28 | 2587.47 | 0.08 | 1.37 | 16.77 | 3.49 (↓80.14%) | 52.86 (↓28.41) |
| **X-Pruner** | 19.76 (2.30×) | 90.92 (3.02×) | 50.61 | 2815.58 | 0.09 | 1.33 | 19.19 | 3.80 (↓78.37%) | 79.11 (↓2.16) |
| **DC-ViT** | 14.76 (3.08×) | 79.66 (3.44×) | 67.75 | 3213.83 | 0.08 | 1.95 | 18.19 | 3.51 (↓80.02%) | 49.18 (↓32.09) |
| **MDP** | 18.30 (2.48×) | 82.79 (3.31×) | 54.64 | 3092.11 | 0.08 | 1.09 | 17.73 | 3.35 (↓80.93%) | 75.73 (↓5.54) |
| **Vulcan** | 15.06 (3.02×) | 99.59 (2.75×) | 66.41 | 2570.61 | 0.08 | 1.40 | 16.96 | 3.26 (↓81.45%) | 88.22 (↑6.95) |

Given that some pruning methods are not hardware-friendly, we compare the computational efficiency of Vulcan with five baselines under different pruning rates. As shown in Table 10, under the same pruning rate, Vulcan-derived models achieve comparable or better inference efficiency, ranking in the top-3 speedups in most cases, demonstrating Vulcan's hardware friendliness. It is worth noting that although DC-ViT attains a significantly higher speedup at a pruning rate of 0.80 by removing part of the MHA modules, the resulting models suffer from overly degraded accuracy, making this trade-off unreasonable.

## J.4 SCALING TO LARGER MODEL AND NEW DOMAIN

Table 11: Performance of Vulcan on larger ViT-L/16 and BERT-Base from the NLP domain. '(FT)' denotes fine-tuning on the class-specific data.

| Methods | ViT-L/16 (ImageNet) | | | BERT-Base (NewsGroups) | | |
|---|---|---|---|---|---|---|
| | T1/25 | T2/25 | T3/25 | T1/2 | T2/4 | T3/8 |
| **Base ViT** | 82.64 | 84.72 | 83.76 | 77.60 | 68.34 | 72.71 |
| **Base ViT (FT)** | 97.52 | 97.28 | 98.16 | 94.43 | 87.95 | 81.52 |
| **Vulcan (0.60)** | 96.72 | 97.10 | 97.58 | 94.56 | 86.41 | 81.66 |
| **Vulcan (0.80)** | 94.54 | 94.73 | 93.75 | 93.16 | 85.68 | 80.19 |

To further demonstrate the scalability of Vulcan, we conduct experiments on both a larger vision model, ViT-L/16, and a language model, BERT-Base, from a different domain. For ViT-L/16, we

derive edge ViTs for the sub-tasks T1–T3 in Table 1. For BERT-Base, we fine-tune the model on the 20 NewsGroups (Lang, 1995) dataset and derive edge ViTs for three sub-tasks with 2, 4, and 8 target classes. As shown in Table 11, Vulcan achieves results comparable to those in Table 1, confirming its strong scalability across both larger architectures and new domains.

## J.5 GENERALIZATION AND ROBUSTNESS OF DERIVED MODELS

Table 12: Accuracy comparison on distribution-shifted datasets between DeiT-Base and Vulcan-derived models for three ImageNet sub-tasks under pruning rates of 0.60 and 0.80. The models derived from DeiT-Base by Vulcan exhibit stronger generalization and robustness on the sub-tasks.

| Methods | ImageNet-V2 | | | ImageNet-R | | |
|---|---|---|---|---|---|---|
| | T1/25 | T2/25 | T3/25 | T1/25 | T2/25 | T3/25 |
| DeiT-Base | 2.40 | 0.40 | 0.00 | 0.00 | 0.00 | 0.00 |
| Vulcan (0.60) | 11.60 | **11.20** | 8.00 | 1.63 | 0.11 | 0.00 |
| Vulcan (0.80) | **12.40** | 7.60 | **10.40** | **2.17** | **2.36** | **0.26** |

To evaluate the generalization and robustness of the edge ViTs derived by Vulcan, we further compare DeiT-Base and its derived edge ViTs on two datasets whose distributions differ from ImageNet:

- ImageNet-V2 (Recht et al., 2019): A re-collection of ImageNet validation data designed to test distribution shifts. It contains images sampled from the same class labels but gathered under different conditions, serving as a benchmark for evaluating model generalization.
- ImageNet-R (Hendrycks et al., 2021): A dataset consisting of various renditions of ImageNet classes (e.g., sketches, paintings, cartoons), which introduces significant appearance variations and is commonly used to assess robustness against domain shifts.

As shown in Table 12, Vulcan can improve the generalization and robustness of base ViTs on sub-tasks to some extent. We attribute this improvement to Vulcan's ability to encourage edge ViTs to focus on specific classes, enhancing their capacity to extract and recognize class-related features. An interesting observation is that models with higher pruning rates exhibit even stronger generalization, suggesting a promising direction for future exploration.

## J.6 VULCAN FOR TRANSFER LEARNING

Vulcan is designed to derive compact models from a pre-trained base ViT for a sub-task $S$, where $S \subseteq Y$ and $Y$ denotes the class set recognized by the base ViT. This corresponds to a closed-domain scenario. In practical applications, however, users often encounter open-domain settings in which the target classes of a sub-task are not contained in $Y$. Such cases fall within the scope of transfer learning. To investigate Vulcan's effectiveness in this context, we employ DeiT-Base pre-trained on ImageNet as the base ViT and evaluate Vulcan-derived models under different pruning rates on three downstream benchmarks: Stanford Cars (Krause et al., 2013), Oxford Flowers-102 (Nilsback & Zisserman, 2008), and Food-101 (Bossard et al., 2014).

Table 13: Overall performance of Vulcan under different pruning rates on downstream tasks. The classes and data distribution of downstream tasks are different from the pre-training data of the base ViT. '(FT)' denotes fine-tuning on the downstream task data.

| Methods | Stanford Cars | Oxford Flowers-102 | Food-101 | Avg |
|---|---|---|---|---|
| DeiT-Base (FT) | 84.02 | 90.58 | 86.81 | 87.14 |
| Vulcan (0.60) | 79.62 | 86.40 | 80.60 | 82.21 |
| Vulcan (0.80) | 66.70 | 78.16 | 74.25 | 73.04 |

As shown in Table 13, at pruning rates of 0.60 and 0.80, the edge ViTs derived by Vulcan achieve 94.34% and 83.82% of the accuracy of models obtained by directly fine-tuning the base ViT. This performance is somewhat lower than the results in Table 1, as the three selected downstream tasks

all belong to fine-grained image recognition (FGVC), which are inherently more challenging and complex than the ImageNet sub-tasks. Nevertheless, the results clearly demonstrate the strong generalizability of Vulcan and its applicability to more difficult open-domain scenarios.

### J.7    WALL-CLOCK COST ANALYSIS

In this section, we present several implementation optimizations and provide a comprehensive analysis of Vulcan's wall-clock cost and scalability.

To begin with, our implementation incorporates several efficiency-oriented design choices:

- **CCNC**: For each FFN block, we concatenate the indices of all clusters into a single tensor, allowing all difference and aggregation operations to be computed in one pass.
- **TNNR**: Instead of performing SVD directly on $W$, we compute the eigen decomposition of $WW^\top$, which is significantly more efficient. We also process all attention heads jointly through batched matrix operations.

With these optimizations, Vulcan does not introduce heavy computation in practice. To quantify the actual overhead, we systematically evaluated how sub-task, pruning rate, and model size affect the time required to compute the CCNC and TNNR regularizers. The wall-clock durations were measured on an NVIDIA A40, and the detailed results are provided in Table 14-16 (all numbers in these tables are measured in seconds).

Table 14: Wall-clock time per training step for computing the CCNC and TNNR regularization terms when Vulcan derives models from DeiT-Base for ImageNet sub-tasks T1/25–T3/25 at a pruning rate of 0.60.

| Loss Term | T1/25 | T2/25 | T3/25 | Avg |
|---|---|---|---|---|
| CCNC | 0.07 | 0.07 | 0.07 | 0.07 |
| TNNR | 0.50 | 0.49 | 0.49 | 0.49 |
| Total | 0.57 | 0.56 | 0.56 | 0.56 |

Table 15: Wall-clock time per training step for computing the CCNC and TNNR regularization terms at different pruning rates when Vulcan derives models from DeiT-Base for the ImageNet sub-task T1/25.

| Loss Term | 0.20 | 0.40 | 0.60 | 0.80 |
|---|---|---|---|---|
| CCNC | 0.06 | 0.07 | 0.07 | 0.06 |
| TNNR | 0.50 | 0.49 | 0.50 | 0.49 |
| Total | 0.56 | 0.56 | 0.57 | 0.55 |

Table 16: Wall-clock time per training step for computing the CCNC and TNNR regularization terms when Vulcan derives models from different-sized base ViTs for ImageNet sub-task T1/25 at a pruning rate of 0.60.

| Loss Term | DeiT-Tiny (5M) | DeiT-Small (22M) | DeiT-Base (86M) | ViT-Large (307M) |
|---|---|---|---|---|
| CCNC | 0.02 | 0.04 | 0.07 | 0.10 |
| TNNR | 0.12 | 0.25 | 0.50 | 1.23 |
| Total | 0.14 | 0.29 | 0.57 | 1.33 |

For example (Table 14), when deriving compact models from DeiT-Base on an NVIDIA A40, Vulcan introduces only 0.56s of extra time per step compared to fine-tuning with the original task loss alone. From these results, combined with Vulcan's design characteristics, we summarize the following observations:

- The cost of CCNC and TNNR is largely insensitive to the choice of sub-task or pruning rate.
- The overhead grows sub-linearly with model size.
- The computation depends only on model weights, and is therefore independent of the batch size.

This implies that in real deployment scenarios, the relative overhead of Vulcan becomes smaller for larger batch sizes and larger models.

## J.8  Trade-off Between Performance and Overhead

**Effects of Batch Size and Learning Rate**. Considering that practical applications often impose varying constraints on memory usage and derivation efficiency, we investigate how batch size (bz) and learning rate (lr)—two critical training configurations directly related to these factors—affect the performance of Vulcan. Specifically, for the three ImageNet sub-tasks T1–T3 defined in Table 1, Vulcan derives models from DeiT-Base under different batch sizes and learning rates. We record memory usage and the accuracy of the derived models across varying batch sizes, as well as the convergence speed of the derivation process under different learning rates.

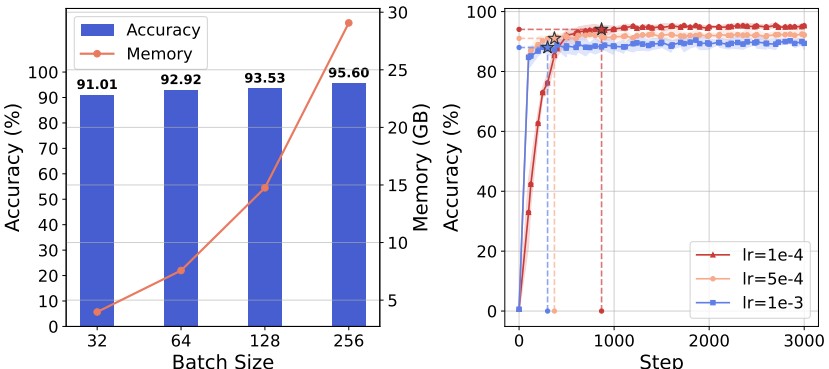

Figure 22: Effects of batch size and learning rate on edge ViTs derived by Vulcan. The asterisks in the right figure mark the step–accuracy points at convergence for each learning rate.

As shown in Figure 22, Vulcan exhibits robustness to batch size: memory usage is reduced by 86.34% with only a 4.59% average drop in accuracy. For the learning rate, convergence is defined as accuracy fluctuations within 0.1% over 100 steps. When the learning rate is set to $5 \times 10^{-4}$ and $10^{-3}$, derivation efficiency improves by 57.14% and 65.44% with accuracy losses of 3.02% and 6.05%, respectively, indicating that higher learning rates can accelerate derivation at a modest accuracy cost.

Table 17: Compatibility of Vulcan with LoRA ($r$=256). The edge ViTs are derived from DeiT-Base with a pruning ratio of 0.60, tailored for the three ImageNet subtasks T1–T3.

| Methods | Memory (GB) | Acc. (%) | | | |
|---|---|---|---|---|---|
| | | T1/25 | T2/25 | T3/25 | Avg |
| **Vulcan** | 1.67 | 95.04 | 96.24 | 95.53 | 95.60 |
| **Vulcan (LoRA)** | 1.10 (↓34.13%) | 88.32 | 89.75 | 89.05 | 89.04 (↓6.56%) |

**Compatibility of Vulcan with LoRA**. In addition, we can also extend Vulcan to LoRA (Hu et al., 2021) to reduce memory usage. We observe that post-training with LoRA requires a relatively larger learning rate to achieve stable convergence. Accordingly, we set the learning rate to $1 \times 10^{-3}$. As shown in Table 17, when adapted to LoRA ($r$=256) with a batch size of 4 and gradient accumulation steps of 64, Vulcan incurs an average accuracy drop of 6.56% while reducing memory consumption by 34.13%. We must acknowledge that LoRA inevitably introduces a degree of accuracy degradation, as optimizing two low-rank matrices alone cannot fully satisfy the constraints associated with $\mathcal{L}_{\text{collapse}}$ and $\mathcal{L}_{\text{rank}}$. Consequently, these losses do not monotonically decrease during the later stages of post-training; instead, they tend to increase monotonically with larger values of $\lambda_1$ and $\lambda_2$. Nevertheless, the derived model still substantially outperforms the base ViT (DeiT-Base), achieving an average improvement of 8.01% across the three sub-tasks.

## J.9  Extreme Model Compression

As discussed in Section 2, Vulcan is orthogonal to other model compression techniques and can be combined with them to enable even lighter deployment. To demonstrate this, we take the edge ViT

$\mathcal{M}_{E/T1}$ derived from DeiT-Base at a pruning rate of 0.80 for the ImageNet sub-task T1/25, and further compress it using knowledge distillation (KD) and INT8 quantization.

Table 18: Results of combining Vulcan with depth pruning, knowledge distillation, and FP16 quantization on ImageNet T1/25. With Vulcan as the core, integrating other compression techniques enables effective lightweighting.

| Methods | Latency (ms) | | Throughput (image/s) | | Storage | #Param | #FLOPs | Acc. |
|---|---|---|---|---|---|---|---|---|
| | Orin NX | RTX 4090 | Orin NX | RTX 4090 | (MB) | (M) | (G) | (%) |
| DeiT-Base | 45.45 | 274.27 | 22.00 | 933.39 | 330.23 | 2.21 | 86.57 | 17.57 |
| Vulcan (T1-0.80) | 15.04 (3.02×) | 99.14 (2.77×) | 66.48 | 2581.97 | 69.48 (↓78.96%) | 18.21 | 3.49 | 92.24 |
| Depth Pruning + KD | 13.16 (3.45×) | 82.24 (3.33×) | 75.96 | 3112.8 | 60.24 (↓81.76%) | 15.79 | 3.06 | 92.32 |
| Quantization (FP16) | 11.35 (4.00×) | 28.34 (9.68×) | 88.07 | 9033.06 | 30.12 (↓90.88%) | 15.79 | 3.06 | 92.32 |

Specifically, we sample 5,000 samples from the T1/25 training set to evaluate the accuracy change after removing each block of $\mathcal{M}_{E/T1}$. Following a greedy strategy, at each step we remove the block that incurs the smallest accuracy drop. Through this depth pruning process, three blocks are removed from $\mathcal{M}_{E/T1}$ to construct a student model for knowledge distillation. Next, we take the DeiT-Base fine-tuned on the T1/25 training set as the teacher model and fine-tune the student model for 5,000 steps using logit-based knowledge distillation with a learning rate of 1e-5. As shown in Table 18, the distilled model achieves an additional 0.08% accuracy gain. We then apply FP16 quantization to the distilled model, further reducing storage cost and improving inference efficiency. Overall, this pipeline reduces the model's storage cost by 90.88%, improves accuracy on the sub-task by 12.40%, and achieves $4.00\times$ and $9.68\times$ inference speedups on Orin NX and RTX 4090, respectively, substantially enhancing the deployability of the model.

# K PSEUDOCODE

---

**Algorithm 1** Vulcan – Class-Specific Model Derivation

---

**Input:** pre-trained base ViT $\mathcal{M}_B$, sub-task $\mathcal{S}$, class-specific dataset $\mathcal{D}_\mathcal{S}$, pruning rate $R$, Lagrange multiplier $\lambda_1, \lambda_2$, penalty parameter $\rho$

**Output:** compact class-specific edge ViT $\mathcal{M}_E$

1: **# Determine the target model architecture**
2: $\{a^{(l,i)}\}_{1 \leq l \leq L, 1 \leq i \leq e_l} \leftarrow \mathcal{M}_B(\mathcal{D}_\mathcal{S})$ $\quad \triangleright$ obtain the activation values of the neurons in $W_1^{(l,h)}$
3: $\{K^{(l)}\}_{1 \leq l \leq L} \leftarrow$ Compute $K^{(l)}$ according to $\{a^{(l,i)}\}_{1 \leq l \leq L, 1 \leq i \leq e_l}$ and Eq. (3)
4: $\{\mathcal{C}_k^{(l)}\}_{1 \leq l \leq L, 1 \leq k \leq K^{(l)}} \leftarrow$ Perform clustering based on $\{K^{(l)}\}_{1 \leq l \leq L}$
5: $\{R_{QK}^{(l)}\}_{1 \leq l \leq L}, \{R_{VO}^{(l)}\}_{1 \leq l \leq L} \leftarrow$ Compute $R_{QK|VO}^{(l)}$ according to Eq. (5)

6: **# Post-training**
7: $\lambda_1, \lambda_2 \leftarrow 0.0, 0.0$
8: step $\leftarrow 0$
9: total step $\leftarrow 6250R^2 + 1250R$ $\quad \triangleright$ empirical relationship between total step and $R$
10: **while** true **do**
11: $\quad$ **for** $t$-th batch $\mathcal{B}_t$ in $\mathcal{D}_\mathcal{S}$ **do**
12: $\quad\quad$ $\{a^{(l,i)}\}_{1 \leq l \leq L, 1 \leq i \leq e_l} \leftarrow$ Forward propagation $\mathcal{M}_B(\mathcal{B}_t)$
13: $\quad\quad$ Update the anchor neuron $\hat{n}_k^{(l)}$ of each cluster $\mathcal{C}_k^{(l)}$ to the neuron with the max activation
14: $\quad\quad$ $\mathcal{L} \leftarrow$ Compute $\mathcal{L}$ according to $\lambda_1, \lambda_2$ and Eq. (8)
15: $\quad\quad$ Backward propagation on $\mathcal{L}$ and update parameters of $\mathcal{M}_B$ using optimizer
16: $\quad\quad$ $\lambda_{1,2} \leftarrow \lambda_{1,2} + \rho \frac{\partial \mathcal{L}}{\partial \lambda_{1,2}}$ $\quad \triangleright$ update the Lagrange multipliers via gradient ascent
17: $\quad\quad$ **if** step $\geq$ total step **then** break
18: $\quad$ **end for**
19: $\quad$ **if** step $\geq$ total step **then** break
20: **end while**

21: **# Pruning**
22: $\mathcal{M}_E = \bigcup_{l=1}^L \{W_1^{(l)}, W_2^{(l)}, \bigcup_{h=1}^{H_l} \{W_Q^{(l,h)}, W_K^{(l,h)}, W_V^{(l,h)}, W_O^{(l,h)}\}\} \leftarrow \mathcal{M}_B$
23: **for** $l$-th block in $\mathcal{M}_E$ **do**
24: $\quad$ Create the weight matrix $W_1^{(l)\prime} \in \mathbb{R}^{K^{(l)} \times d}, W_2^{(l)\prime} \in \mathbb{R}^{d \times K^{(l)}}$
25: $\quad$ $W_1^{(l)\prime}, W_2^{(l)\prime} \leftarrow$ Pruning according to Eq. (9)
26: $\quad$ **for** $h$-th head in $l$-th block in $\mathcal{M}_E$ **do**
27: $\quad\quad$ $q_l', v_l' \leftarrow \lfloor q_l(1 - R_{QK}^{(l)}) \rfloor, \lfloor v_l(1 - R_{VO}^{(l)}) \rfloor$
28: $\quad\quad$ Create the weight matrix $W_{Q|K}^{(l,h)\prime} \in \mathbb{R}^{q_l' \times d}, W_V^{(l,h)\prime} \in \mathbb{R}^{v_l' \times d}, W_O^{(l,h)\prime} \in \mathbb{R}^{d \times v_l'}$
29: $\quad\quad$ $W_Q^{(l,h)\prime}, W_K^{(l,h)\prime}, W_V^{(l,h)\prime}, W_O^{(l,h)\prime} \leftarrow$ Pruning according to Eq. (10) and Eq. (11)
30: $\quad$ **end for**
31: **end for**
32: $\mathcal{M}_E \leftarrow \bigcup_{l=1}^L \{W_1^{(l)\prime}, W_2^{(l)\prime}, \bigcup_{h=1}^{H_l} \{W_Q^{(l,h)\prime}, W_K^{(l,h)\prime}, W_V^{(l,h)\prime}, W_O^{(l,h)\prime}\}\}$
33: **return** $\mathcal{M}_E$

---

## L   NOTATION TABLE

Table 19: Notation Table1.

| Symbol | Description |
|---|---|
| $N$ | Number of patch tokens in the sequence, obtained by partitioning the input image or signal into fixed-size patches. |
| $d$ | Embedding dimension of each patch token, indicating the length of the feature vector associated with every token. |
| $\mathbf{X}$ | Patch token sequence $\mathbf{X} \in \mathbb{R}^{N \times d}$, consisting of $N$ tokens, each represented by a $d$-dimensional embedding vector. |
| $\text{MHA}^{(l)}(\cdot)$ | Multi-Head Attention module in the $l$-th ViT block. |
| $\text{Attn}^{(l,h)}(\cdot)$ | The attention operation of the $h$-th head in the $l$-th block. |
| $H_l$ | The number of attention heads in the $l$-th block. |
| $q_l$ | Dimensionality of the query and key vectors (query-key dimension) in the $l$-th block. |
| $v_l$ | Dimensionality of the value vectors (value-output dimension) in the $l$-th block. |
| $W_Q^{(l,h)}$ | Query projection matrix $W_Q^{(l,h)} \in \mathbb{R}^{q_l \times d}$ for the $h$-th attention head in the $l$-th block. |
| $W_K^{(l,h)}$ | Key projection matrix $W_K^{(l,h)} \in \mathbb{R}^{q_l \times d}$ for the $h$-th attention head in the $l$-th block. |
| $W_V^{(l,h)}$ | Value projection matrix $W_V^{(l,h)} \in \mathbb{R}^{v_l \times d}$ for the $h$-th attention head in the $l$-th block. |
| $W_O^{(l,h)}$ | Output projection matrix $W_O^{(l,h)} \in \mathbb{R}^{d \times v_l}$ for the $h$-th attention head in the $l$-th block. |
| $\text{FFN}^{(l)}(\cdot)$ | Feed-Forward Network module in the $l$-th ViT block. |
| $e_l$ | Intermediate/expansion dimension of the FFN in $l$-th block, corresponding to the number of neurons in the $\text{FFN}^{(l)}$. |
| $W_1^{(l)}$ | First FFN projection matrix $W_1^{(l)} \in \mathbb{R}^{e_l \times d}$ in the $l$-th block. |
| $W_2^{(l)}$ | Second FFN projection matrix $W_2^{(l)} \in \mathbb{R}^{d \times e_l}$ in the $l$-th block. |
| $W[i]$ | The $i$-th row of $W$. |
| $W[:i]$ | The $i$-th column of $W$. |
| $W[:,i]$ | Submatrix of $W$ consisting of its first $i$ rows (row slice). |
| $W[:,:i]$ | Submatrix of $W$ consisting of its first $i$ columns (column slice). |
| $\sigma(\cdot)$ | Nonlinear activation function applied within the FFN module (e.g., GELU). |
| $\otimes$ | Outer product operator. |
| $n_1^{(l,i)}$ | The $i$-th neuron $n_1^{(l,i)} = W_1^{(l)}[i] \in \mathbb{R}^d$ (row vector) in $W_1^{(l)}$. |
| $n_2^{(l,i)}$ | The $i$-th neuron $n_2^{(l,i)} = W_2^{(l)}[:,i] \in \mathbb{R}^d$ (column vector) in $W_2^{(l)}$. |
| $a^{(l,i)}$ | Activation value $a^{(l,i)} = \sum_{j=1}^{N} \sigma\left( \langle \mathbf{X}[j], n_1^{(l,i)} \rangle \right)$ of the $i$-th neuron $n_1^{(l,i)}$ in the FFN of the $l$-th block. |
| $R$ | Overall pruning rate $R \in (0,1)$. |
| $\Phi(\mathcal{A}, k)$ | Function that returns the $k$-th smallest element in the set $\mathcal{A}$. |
| $\mathbb{I}(\cdot)$ | Indicator function that outputs 1 if its condition is true and 0 otherwise. |
| $K^{(l)}$ | Number of clusters formed in the FFN of the $l$-th block. |
| $\mathcal{C}_k^{(l)}$ | The $k$-th cluster $\mathcal{C}_k^{(l)} = \{n_k^{(l,i)}\}_{i=1}^{|\mathcal{C}_k^{(l)}|}$ in the FFN of the $l$ the block. |
| $a_k^{(l,i)}$ | The activation value of the $i$-th neuron within cluster $\mathcal{C}_k^{(l)}$. |
| $[M]$ | The collection of the first $M$ positive integers $\{1, 2, 3, \cdots, N\}$. |

Table 20: Notation Table2.

| Symbol | Description |
|---|---|
| $i^*$ | The index of the neuron within cluster $\mathcal{C}_k^{(l)}$ that attains the maximum activation value, i.e., $i^* = \arg\max\limits_{i \in [|\mathcal{C}_k^{(l)}|]} a_k^{(l,i)}$ |
| $n_k^{(l,i)}$ | The $i$-th neuron $n_k^{(l,i)} = n_1^{(l,j)} \in \mathbb{R}^d$ in cluster $\mathcal{C}_k^{(l)}$. |
| $\hat{n}_k^{(l)}$ | The anchor neuron $\hat{n}_k^{(l)} = \hat{n}_k^{(l,i^*)}$ of cluster $\mathcal{C}_k^{(l)}$. |
| $\nu_k^{(l,i)}$ | The value associated with the $i$-th neuron in cluster $\mathcal{C}_k^{(l)}$, representing either its weight vector $n_k^{(l,i)}$ or activation $a_k^{(l,i)}$. |
| $\hat{\nu}_k^{(l)}$ | The value associated with the anchor neuron in cluster $\mathcal{C}_k^{(l)}$, representing either its weight vector $n_k^{(l,i^*)}$ or activation $a_k^{(l,i^*)}$. |
| $W_{QK}^{(l,h)}$ | The query–key composite matrix for the $h$-th head in the $l$-th block, defined as $W_{QK}^{(l,h)} = W_Q^{(l,h)\top} W_K^{(l,h)} \in \mathbb{R}^{d \times d}$. |
| $W_{VO}^{(l,h)}$ | The value–output composite matrix for the $h$-th head in the $l$-th block, defined as $W_{VO}^{(l,h)} = W_V^{(l,h)\top} W_O^{(l,h)\top} \in \mathbb{R}^{d \times d}$. |
| $\mathcal{E}(W)$ | The effective rank of matrix $W$. |
| $r_{QK}^{(l)}$ | Dimension-level effective rank of the query–key composite in the $l$-th block, computed as the sum over all $H_l$ heads of the effective ranks $\mathcal{E}(W_{QK}^{(l,h)})$. |
| $r_{VO}^{(l)}$ | Dimension-level effective rank of the value–output composite in the $l$-th block, computed as the sum over all $H_l$ heads of the effective ranks $\mathcal{E}(W_{VO}^{(l,h)})$. |
| $R^{(l)}$ | Pruning rate of the MHA module in the $l$-th block. |
| $R_{QK}^{(l)}$ | Pruning rate of the query–key (QK) dimension in the MHA module of the $l$-th block. |
| $R_{VO}^{(l)}$ | Pruning rate of the value–output (VO) dimension in the MHA module of the $l$-th block. |
| $\sigma_Q^{(l,h,i)}$ | The $i$-th singular value of $W_Q^{(l,h)}$. |
| $\sigma_V^{(l,h,i)}$ | The $i$-th singular value of $W_V^{(l,h)}$. |
| $U_{QK}^{(l,h)}, \Sigma_{QK}^{(l,h)}, V_{QK}^{(l,h)}$ | The left singular vectors $U_{QK}^{(l,h)} \in \mathbb{R}^{d \times d}$, singular values $\Sigma_{QK}^{(l,h)} \in \mathbb{R}^{d \times d}$, and right singular vectors $V_{QK}^{(l,h)} \in \mathbb{R}^{d \times d}$ of the $W_{QK}^{(l,h)}$ obtained by its singular value decomposition. |
| $U_{VO}^{(l,h)}, \Sigma_{VO}^{(l,h)}, V_{VO}^{(l,h)}$ | The left singular vectors $U_{VO}^{(l,h)} \in \mathbb{R}^{d \times d}$, singular values $\Sigma_{VO}^{(l,h)} \in \mathbb{R}^{d \times d}$, and right singular vectors $V_{VO}^{(l,h)} \in \mathbb{R}^{d \times d}$ of the $W_{VO}^{(l,h)}$ obtained by its singular value decomposition. |

