# OpenReview forum: "Vulcan: Crafting Compact Class-Specific Vision Transformers For Edge Intelligence"
_ICLR.cc/2026/Conference — ICLR 2026 Poster_

### Official Review · Reviewer_Z9iw · 2025-10-31

**Soundness:** 3
**Presentation:** 2
**Contribution:** 3
**Rating:** 6
**Confidence:** 4

**Summary:**

The paper introduces Vulcan, a pruning-oriented post-training framework for deriving class-specific compact Vision Transformers for edge deployment. The key insight comes from that FFN neurons encode class-specific knowledge, while MHA layers capture class-agnostic patterns. Based on this, Vulcan proposes two complementary techniques:
(1) CCNC clusters FFN neurons by activation and collapses them toward anchor neurons, and
(2) TNNR, which induces low-rank structure in MHA weights to enable near-lossless SVD pruning.
A unified train-then-prune optimization via augmented Lagrangian enforces these redundancies.

**Strengths:**

1. Edge deployment with individualization is an interesting and important problem in edge cloud cooperation, and the authors provide a practical and effective pipeline solution.
2. The insight on knowledge disentanglement between FFN and MHA modules is convincing and well-supported by visualization and interpretability experiments.
3. Experimental evaluation is comprehensive, covering multiple architectures (DeiT, Swin), tasks (classification, detection, segmentation), and hardware setups (Jetson, RTX 4090).
4. The ablation studies and visual analyses are comprehensive.

**Weaknesses:**

1. The writeup is somewhat complicated. Although the steps and concepts are strictly formalized, there is little intuition or explanation, making the formulations and notations hard to follow. For example, in the term $a^{(l,i)} = \sum_i \sigma(X[i] \cdot n^{(l,i)}_1)$ , it seems  $i$ is summed out on the right-hand side, but the left-hand side still includes $i$ , which is confusing.
2. While conceptually clear, the method section is dense and overly mathematical, which makes it difficult to grasp the main ideas of the augmented Lagrangian optimization.
3. The methodology feels more like a collection of engineering techniques combined into one pipeline rather than a focused research exploration, with too many tricks applied at once.

**Questions:**

1. Could CCNC overfit to small subtasks if activations are dominated by a few classes?
2. How sensitive is the neuron clustering step to dataset imbalance or class overlap?

---

> ### Author Response · Authors · 2025-11-19
> **Response 1/2**
>
> We humbly appreciate your thoughtful feedback on our work! For a better understanding of our rebuttal and revision, we have summarized your key concerns and our responses as follows:
>
> > **W1**: The writeup is somewhat complicated. Although the steps and concepts are strictly formalized, there is little intuition or explanation, making the formulations and notations hard to follow. For example, in the term $a^{(l,i)}=\sum_i\sigma(X[i]\cdot n_1^{(l,i)})$, it seems $i$ is summed out on the right-hand side, but the left-hand side still includes $i$ , which is confusing.
>
> Thank you for pointing out the confusion in the activation-value formulation. We apologize for the misunderstanding. The correct expression should be $a^{(l,i)}=\sum_j\sigma(X[j]\cdot n_1^{(l,i)})$, where the summation index is $j$ rather than $i$, and "$\cdot$" denotes the inner product between two vectors. We have corrected this in the revised version.
>
> More broadly, we appreciate the feedback regarding the dense use of symbols and equations in our methods section. We apologize for the difficulty and have taken several steps to improve the presentation in the revision. In particular, we ensured that all symbols are clearly defined upon first use and added consolidated notation tables (Tables 15 and 16 in Appendix L) for easy reference. We also optimized several formulas to avoid potential misunderstandings, including the activation-value computation above and the definition of $i^*$ in Equation (4). We hope that these improvements enhance the readability and clarity of the paper while preserving the necessary theoretical rigor.
>
> To further facilitate understanding, we provide a brief roadmap of the equations used in the paper:
>
> - **Determining the target model architecture**. Before post-training, Vulcan must determine the target architecture corresponding to pruning rate $R$. Equations (3), (5), and (6) compute the three pruned dimensions.
> - **Regularization terms**. Vulcan introduces CCNC and TNNR through the regularizers in Equations (4) and (7), which respectively encourage duplicated neurons in FFN modules and low-rank structure in MHA modules.
> - **Overall loss function**. Equation (8) presents the final optimization objective of Vulcan, where the first- and second-order constraints derived from Equations (4) and (7) are combined with the original task loss.
> - **Pruning procedure**. Equations (9)–(11) describe Vulcan’s pruning process after redundancy has been introduced.
>
> We hope this structured explanation helps clarify the mathematical flow of the method. We sincerely appreciate the reviewer’s constructive feedback.
>
> > **W2**: While conceptually clear, the method section is dense and overly mathematical, which makes it difficult to grasp the main ideas of the augmented Lagrangian optimization.
>
> We appreciate the reviewer’s concern and are happy to provide a concise and intuitive explanation of the augmented Lagrangian optimization used in our method. The regularization terms corresponding to CCNC and TNNR can be viewed as two constraints, and Vulcan’s goal during post-training is to satisfy these constraints so that the subsequent pruning can be nearly lossless. The augmented Lagrangian framework is a classical approach for handling such constrained optimization problems. It augments the first-order penalty terms (Equations (4) and (7)) with an additional quadratic penalty, and introduces two learnable Lagrange multipliers, $\lambda_1$ and $\lambda_2$, associated with the respective constraints. During post-training, these multipliers gradually increase to strengthen the enforcement of the constraints, ensuring that the model not only minimizes the task objective but also satisfies the structural conditions required for near-lossless pruning.
>
> We plan to add a brief explanation of this process to Section 3.4 in the main paper to further help readers follow the intuition behind the optimization. Thank you for the helpful suggestion.

---

> ### Author Response · Authors · 2025-11-19
> **Response 2/2**
>
> > **W3**: The methodology feels more like a collection of engineering techniques combined into one pipeline rather than a focused research exploration, with too many tricks applied at once.
>
> We apologize if our presentation gave the impression that the method is a collection of loosely connected techniques. In fact, each component of Vulcan is tightly integrated and derived from a coherent design rationale. We summarize the overall logic as follows:
>
> - **Adaptively determining the target model architecture**. Before constructing the CCNC and TNNR regularizers, Vulcan must determine the pruned architecture. Figure 5 and Appendix D provide the empirical basis for our choices of MHA pruning dimensions and pruning configurations. Appendix H shows the advantage of determining the pruned FFN intermediate dimension adaptively based on activation statistics rather than fixing it manually.
> - **Designing regularizers that introduce the required redundancy**. Once the target architecture is fixed, we construct principled regularizers that explicitly induce redundancy in FFN and MHA modules. For ​**CCNC**​, the interpretability of FFN neurons (Figure 3) motivates using activations to guide neuron collapse, which encourages the model to focus on class-specific patterns. Figure 10 shows that applying the collapse constraint on weights is most effective. For ​**TNNR**​, Figure 4 demonstrates that SVD preserves useful knowledge significantly better than score-based methods, a property theoretically supported by the Eckart–Young theorem. Therefore, we intentionally drive part of the singular values toward zero to enable lossless SVD-based pruning.
> - **Post-training and pruning**. Our goal is to ensure that, after post-training, the weights to be pruned contain ​*no useful knowledge*​, so that pruning results in ​*nearly no accuracy loss*​. This requires both CCNC and TNNR constraints to be strictly satisfied. To achieve this, we optimize the model using the augmented Lagrangian framework, where gradually increasing Lagrange multipliers enforce the constraints tightly during training.
>
> We hope this clarification makes the underlying coherence of Vulcan’s design more apparent.
>
> > **Q1**: Could CCNC overfit to small sub-tasks if activations are dominated by a few classes?
>
> We report Vulcan’s performance on the *smallest-scale sub-tasks* across several datasets. For recognition tasks, we use DeiT-Base as the base ViT; for detection and segmentation tasks, we use Fast/Mask R-CNN (Swin-T) as the base model. The results are shown in the table below:
>
> | Method         | ImageNet (\|$\mathcal{S}$\|=2) | CIFAR100 (\|$\mathcal{S}$\|=2) | CIFAR10 (\|$\mathcal{S}$\|=2) | COCO (\|$\mathcal{S}$\|=1)  |
> |----------------|--------------------|--------------------|-------------------|-----------------|
> | Base ViT       | 76.00              | 80.50              | 98.05             | 66.5/68.6       |
> | Vulcan (0.60)  | 100.00             | 97.50              | 99.05             | 68.2/70.8       |
> | Vulcan (0.80)  | 100.00             | 94.00              | 98.45             | 57.6/61.8       |
>
> We observe that Vulcan does not exhibit overfitting under CCNC when the sub-task is extremely small.
>
> > **Q2**: How sensitive is the neuron clustering step to dataset imbalance or class overlap?
>
> To evaluate the sensitivity of the neuron-collapse step under dataset imbalance, we construct long-tailed variants of each sub-task dataset $\mathcal{D_S}$ using the following transformation:
>
> $$
> n_i = n\times IF^{\frac{1-i}{|\mathcal{S}|-1}}
> $$
>
> where $n$ is the per-class count in the balanced set, $n_i$ is the count for the $i$-th class after imbalance, $|\mathcal{S}|$ is the number of classes in the sub-task, and **IF** is the imbalance factor (IF=1 means balanced).
>
> We derive class-specific models from DeiT-Base for an ImageNet sub-task with 25 random classes under IF = 1, 10, 20, and 50. The results are shown in the table below:
>
> | Method         | IF=1  | IF=10 | IF=20 | IF=50 |
> |----------------|-------|-------|-------|-------|
> | DeiT-Base      |   79.92    | 79.92 |   79.92    |   79.92    |
> | DeiT-Base (FT) | 96.40 | 94.64 | 93.00 | 91.44 |
> | Vulcan (0.60)  | 95.04 | 92.00 | 90.80 | 88.32 |
> | Vulcan (0.80)  | 92.24 | 88.76 | 84.84 | 81.32 |
>
> We observe that the larger the imbalance factor, the lower the performance of the derived model. This behavior is expected because fine-tuning a ViT directly on the same imbalanced datasets also leads to notable accuracy degradation. Importantly, applying techniques such as data augmentation or class re-weighting can mitigate this effect to a significant extent, and Vulcan is fully compatible with these techniques.
>
> We thank the reviewer again for the constructive feedback. We sincerely hope that our clarifications and additional experiments address your concerns, and we would be very glad to discuss them further. We remain very open to any additional questions or suggestions.

---

> ### Author Response · Authors · 2025-11-26
> **Rebuttal follow-up**
>
> Dear Reviewer Z9iw,
>
> I hope this message finds you well. We would like to kindly follow up regarding our rebuttal submission. As the discussion phase will conclude in about a week, we sincerely value any additional feedback you may have. In particular, we would greatly appreciate knowing whether our responses have addressed your concerns or if there are remaining questions that we could further clarify. We would be very glad to provide any additional information promptly.
>
> Thank you again for your time and for your thoughtful reviews.

---

### Official Review · Reviewer_J15G · 2025-10-31

**Soundness:** 3
**Presentation:** 3
**Contribution:** 3
**Rating:** 6
**Confidence:** 3

**Summary:**

This paper focuses on the class-specific requirements of deploying ViT on edge devices. Many edge scenarios require only a small number of target classes to be recognized, but the large amount of irrelevant class knowledge carried by pre-trained ViT consumes resources and can interfere with target class performance. The authors claim that by analyzing the knowledge distribution of ViT modules, they find that FFNs favor class-specific knowledge, while MHAs favor class-independent patterns. Based on this, they propose a train-then-prune post-training framework Vulcan. Vulcan introduces redundancy into ViTs deliberately by collapsing FFN neurons onto those with the highest class-specific activations and by enforcing low-rankness in MHA weights.

**Strengths:**

1.	The problem studied in this paper is meaningful, namely, achieving lightweight model deployment at the edge. The authors also have a good insight that not all types of information are useful.
2.	CCNC clusters FFN neurons by "activating the highest anchor point", and TNNR adaptively distributes pruning rates by effective rank. Both are consistent with the observation of "FFN class specific/MHA class independent"; they are unified in the augmented Lagrangian objective, which facilitates direct implementation of structured pruning.
3.	The authors conducted sufficient experiments to demonstrate the effectiveness of the proposed framework.

**Weaknesses:**

1.	The paper uses visualization and the phenomenon that "data-free SVD is better than score-based pruning" to support the claim that "MHA is more class-agnostic ", but lacks more rigorous verification. Can some other models or datasets be used to further support this observation?
2.	TNNR introduces additional computations of singular values/effective ranks, which may incur significant overhead; the paper does not provide data on wall clock duration and scalability for large models/big data scenarios.
3.	The author can provide a rough sketch of the theoretical analysis in the main paper to improve the completeness of the entire work.

**Questions:**

1.	The paper uses visualization and the phenomenon that "data-free SVD is better than score-based pruning" to support the claim that "MHA is more class-agnostic ", but lacks more rigorous verification. Can some other models or datasets be used to further support this observation?
2.	Can the authors report the relative training time/GPU memory overhead of TNNR (relative to fine-tuning only the task loss) on different platforms/datasets?
3.	The random subtask is only divided into 3 categories. Can you provide more mean ± variance or confidence intervals of the random divisions, especially under high shear rates of 0.8/0.9, to prove stability?

---

> ### Author Response · Authors · 2025-11-19
> **Response 1/3**
>
> We humbly appreciate your thoughtful feedback on our work! For a better understanding of our rebuttal and revision, we have summarized your key concerns and our responses as follows:
>
> > **W1**: The paper uses visualization and the phenomenon that "data-free SVD is better than score-based pruning" to support the claim that "MHA is more class-agnostic ", but lacks more rigorous verification. Can some other models or datasets be used to further support this observation?
>
> We appreciate the reviewer’s suggestion to further validate our observation that FFN modules encode *class-specific* knowledge while MHA modules capture *class-agnostic* patterns. In response, we conducted additional experiments on four representative model–dataset pairs to validate this observation:
>
> - DeiT-Tiny / CIFAR-10
> - DeiT-Small / CIFAR-100
> - ViT-Large / ImageNet-1K
> - Mask R-CNN (Swin-T) / COCO
>
> We extended the FFN-activation visualizations in the revised PDF (Appendix B, Figures 15–18). Across all ViT variants and datasets, FFN neurons consistently exhibit highly interpretable activation patterns, reinforcing that FFNs store class-specific knowledge. To further examine the class-agnostic nature of MHA, we compared data-free SVD with three score-based pruning methods (activation, gradient, and Taylor approximation) under a dimensional (query-key: QK, value-output: VO) pruning rate of 0.60. The results are shown in the table below:
>
> | Model & Dataset                 | Method            | QK          | VO          |
> |---------------------------------|-------------------|-------------|-------------|
> | DeiT-Tiny (C10)   | Activation        | 52.57       | 21.01       |
> |                                 | Gradient          | 47.96       | 36.57       |
> |                                 | Taylor Expansion  | 79.09       | 36.25       |
> |                                 | **SVD**               | **93.78**       | **61.62**       |
> | DeiT-Small (C100)  | Activation        | 32.52       | 2.88        |
> |                                 | Gradient          | 30.10       | 4.62        |
> |                                 | Taylor Expansion  | 44.91       | 3.84        |
> |                                 | **SVD**               | **81.00**       | **51.04**       |
> | ViT-Large (IN) | Activation        | 42.46       | 19.29       |
> |                                 | Gradient          | 46.04       | 34.49       |
> |                                 | Taylor Expansion  | 56.84       | 33.01       |
> |                                 | **SVD**               | **79.47**       | **72.87**       |
> | Mask R-CNN (COCO)  | Activation        | 27.3/26.9   | 0.0/0.0     |
> |                                 | Gradient          | 27.4/26.9   | 0.0/0.0     |
> |                                 | Taylor Expansion  | 26.7/26.5   | 0.0/0.0     |
> |                                 |**SVD**              | **38.3/36.3**   | **6.4/6.5**     |
>
> Across all settings, ​SVD consistently and significantly outperforms all score-based criteria​, providing strong evidence that MHA weights encode knowledge that is largely ​independent of specific classes or input examples​. We believe this behavior stems from the intrinsic structure of attention computation. In particular, the two consecutive linear transformations in attention ($W_Q^\top W_K$, $W_V^\top W_O^\top$) operate in a way that is not directly dependent on the identity of the input patch tokens. This property encourages the model to store more generic and reusable knowledge in the query–key and value–output subspaces during pre-training.
>
> The effectiveness of SVD is also theoretically supported, as discussed in Appendix E. According to the Eckart-Young theorem, SVD yields the minimal Frobenius-norm change to $W_Q^\top W_K$ and $W_V^\top W_O^\top$ after pruning. This enables SVD to preserve class-agnostic knowledge in these two subspaces as much as possible.

---

> ### Author Response · Authors · 2025-11-19
> **Response 2/3**
>
> > **W2**: TNNR introduces additional computations of singular values/effective ranks, which may incur significant overhead; the paper does not provide data on wall clock duration and scalability for large models/big data scenarios.
>
> Below we clarify several implementation optimizations and provide a comprehensive analysis of Vulcan’s wall-clock cost and scalability.
>
> To begin with, our implementation incorporates several efficiency-oriented design choices:
>
> - CCNC: For each FFN block, we concatenate the indices of all clusters into a single tensor, allowing all difference and aggregation operations to be computed in one pass.
> - TNNR: Instead of performing SVD directly on $W$, we compute the eigen decomposition of $WW^\top$, which is significantly more efficient. We also process all attention heads jointly through batched matrix operations.
>
> With these optimizations, Vulcan does not introduce heavy computation in practice. To quantify the actual overhead, we systematically evaluated how ​sub-task​, ​pruning rate​, and model size affect the time required to compute the CCNC and TNNR regularizers. The wall-clock durations were measured on an NVIDIA A40, and the detailed results are provided in the three tables below (all numbers in the table are measured in seconds):
>
> - Model: DeiT-Base; Pruning Rate: 0.60
>
>  | Loss  / Sub-task                    | T1/25 | T2/25 | T3/25 | Avg  |
> |---------------------------|-------|-------|-------|------|
> | CCNC      | 0.07  | 0.07  | 0.07  | 0.07 |
> |TNNR         | 0.50  | 0.49  | 0.49  | 0.49 |
> | Total       | 0.57  | 0.56  | 0.56  | 0.56 |
>
>
> - Model: DeiT-Base; Sub-task: T1/25
>
> | Loss   / Pruning Rate              | 0.20 | 0.40 | 0.60 | 0.80 |
> |----------------------|------|------|------|------|
> | CCNC | 0.06 | 0.07 | 0.07 | 0.06 |
> | TNNR     | 0.50 | 0.49 | 0.50 | 0.49 |
> | Total | 0.56 | 0.56 | 0.57 | 0.55 |
>
>
> - Pruning Rate: 0.60; Sub-task: T1/25
>
>  | Loss  / Model                 | DeiT-Tiny (5M) | DeiT-Small (22M) | DeiT-Base (86M) | ViT-Large (307M) |
> |---------------------------|----------------|-------------------|------------------|-------------------|
> | CCNC      | 0.02           | 0.04              | 0.07             | 0.10              |
> | TNNR          | 0.12           | 0.25              | 0.50             | 1.23              |
> | Total                     | 0.14           | 0.29              | 0.57             | 1.33              |
>
>
>
> For example, when deriving compact models from DeiT-Base on an NVIDIA A40, Vulcan introduces only **0.56s** of extra time per step compared to fine-tuning with the original task loss alone. From these results, combined with Vulcan’s design characteristics, we summarize the following observations:
>
> - The cost of CCNC and TNNR is largely insensitive to the choice of sub-task or pruning rate.
> - The overhead grows sub-linearly with model size.
> - ​The computation depends only on model weights​, and is therefore ​independent of the batch size​.
>
> This implies that in real deployment scenarios, the *relative* overhead of Vulcan becomes smaller for larger batch sizes and larger models.
>
> Furthermore, we provide end-to-end wall-clock duration and memory overhead for Vulcan's post-training on multiple platforms and datasets, and compare them with fine-tuning using only original task loss. The results are shown in the table below:
>
> Model: DeiT-Base; Pruning Rate = 0.60; Post-training Steps: 3000; Batch Size: 256 (A40), 128 (4090)
>
> - Sub-task: T1/25 (ImageNet)
>
> |Setting | Training Time (min) | GPU Memory (GB) |
> | --- | --- | --- |
> | Task Loss-Only (A40) | 88.43 | 29.53 |
> | Vulcan (A40) | 120.17 | 30.05 |
> | Task Loss-Only (4090) | 23.22 | 15.42 |
> | Vulcan (4090) | 40.88 | 15.73 |
>
>
> - Sub-task: T1/10 (CIFAR100)
>
> |Setting | Training Time (min) | GPU Memory (GB) |
> | --- | --- | --- |
> | Task Loss-Only (A40) | 89.12 | 29.53 |
> | Vulcan (A40) | 120.25 | 30.05 |
> | Task Loss-Only (4090) | 23.18 | 15.42 |
> | Vulcan (4090) | 41.25 | 15.73 |
>
>
>
> While Vulcan inevitably incurs additional cost because it not only adapts the model to the sub-task but also actively constructs redundancy for near-lossless pruning, we believe this overhead is cost-effective. The derived lightweight models can subsequently run on edge devices with significantly lower compute and memory footprints for long-term real-time inference, making the one-time overhead of Vulcan a worthwhile trade-off.

---

> ### Author Response · Authors · 2025-11-19
> **Response 3/3**
>
> > **W3**: The author can provide a rough sketch of the theoretical analysis in the main paper to improve the completeness of the entire work.
>
> We thank the reviewer for this helpful suggestion. In the revised version, we plan to move the proof of near-lossless pruning from Appendix G into the main paper as a new Section 3.5, so that the theoretical analysis becomes more accessible and improves the overall completeness of the work. If there are additional theoretical components the reviewer would like to see included, we would be very happy to provide them.
>
> > Q1: The paper uses visualization and the phenomenon that "data-free SVD is better than score-based pruning" to support the claim that "MHA is more class-agnostic ", but lacks more rigorous verification. Can some other models or datasets be used to further support this observation?
>
> Please see our response to **W1**.
>
> > Q2: Can the authors report the relative training time/GPU memory overhead of TNNR (relative to fine-tuning only the task loss) on different platforms/datasets?
>
> Please see our response to **W2**.
>
> > Q3: The random sub-task is only divided into 3 categories. Can you provide more mean ± variance or confidence intervals of the random divisions, especially under high shear rates of 0.8/0.9, to prove stability?
>
> To evaluate the robustness of Vulcan under different sub-task configurations, we randomly sampled 10 sub-tasks from ImageNet for each target sub-task size. The table below reports the mean ± standard deviation of the derived models from DeiT-Base at pruning rates 0.80 and ​0.90​:
>
> | Method        | \|$\mathcal{S}$\|=25         | \|$\mathcal{S}$\|=50         | \|$\mathcal{S}$\|=100        |
> |---------------|------------------|------------------|------------------|
> | DeiT-Base     | 82.01 (±2.21)    | 81.96 (±2.40)    | 81.36 (±2.37)    |
> | Vulcan (0.80) | 93.60 (±1.44)    | 88.67 (±0.69)    | 83.36 (±1.42)    |
> | Vulcan (0.90) | 91.27 (±1.92)    | 84.54 (±1.39)    | 78.31 (±1.21)    |
>
> We thank the reviewer again for the constructive feedback. We sincerely hope that our clarifications and additional experiments address your concerns, and we would be very glad to discuss them further. We remain very open to any additional questions or suggestions.

---

> ### Author Response · Authors · 2025-11-26
> **Rebuttal follow-up**
>
> Dear Reviewer J15G,
>
> I hope this message finds you well. We would like to kindly follow up regarding our rebuttal submission. As the discussion phase will conclude in about a week, we sincerely value any additional feedback you may have. In particular, we would greatly appreciate knowing whether our responses have addressed your concerns or if there are remaining questions that we could further clarify. We would be very glad to provide any additional information promptly.
>
> Thank you again for your time and for your thoughtful reviews.

---

> > ### Comment · Reviewer_J15G · 2025-11-27
> >
> > Thank you for the detailed response, I tend to keep my positive score.

---

### Official Review · Reviewer_QM9G · 2025-11-01

**Soundness:** 2
**Presentation:** 3
**Contribution:** 3
**Rating:** 4
**Confidence:** 3

**Summary:**

The paper proposes Vulcan, a framework for class-specific pruning of pre-trained Vision Transformers (ViTs). In ViTs, the FFN layers primarily capture class-related features, while the MHA layers tend to preserve class-agnostic information. Based on this property, Vulcan applies different compression strategies to the two components: redundant neurons in FFN layers are merged through neuron clustering and collapse, and attention matrices in MHA layers are regularized with truncated nuclear norm (TNNR) to induce low-rank structures. After a post-training stage, the framework directly derives a compact ViT specialized for a subset of target classes. Experiments on multiple datasets demonstrate that Vulcan improves classification accuracy under high pruning ratios and achieves notable speedups.

**Strengths:**

1. The experimental evaluation is extensive, covering multiple datasets (ImageNet, CIFAR-10/100, COCO) and tasks (classification and detection). Results are reported under various pruning ratios with comprehensive metrics including accuracy, FLOPs, parameter count, and hardware latency.
2. Implementation details are clearly presented, making the method reproducible. The post-training behavior, hyperparameter sensitivity, and neuron visualization provide additional insights into the framework.

**Weaknesses:**

1. The overall idea builds on the known observation that FFN layers in ViTs capture class-related information, and both the neuron clustering and low-rank decomposition techniques are well-established; thus, the level of novelty is limited.
2. The application scenario involves only a small number of target classes, which could often be addressed by lightweight models trained on smaller datasets, leaving the necessity of this approach unclear.
3. The method section contains an excessive amount of symbols and equations, making the presentation somewhat dense and difficult to follow.

**Questions:**

1. It is not specified whether the baseline models were also retrained on the same class subsets—could this affect the fairness of the comparisons?
2. The paper does not compare Vulcan with lightweight models designed for limited-class scenarios; would Vulcan still provide a clear advantage under such settings?
3. The FFN collapse and MHA low-rank compression losses are directly added together—could there be potential interactions between the two, and is such joint optimization theoretically justified?

---

> ### Author Response · Authors · 2025-11-19
> **Response 1/3**
>
> We humbly appreciate your thoughtful feedback on our work! For a better understanding of our rebuttal and revision, we have summarized your key concerns and our responses as follows:
>
> > **W1**: The overall idea builds on the known observation that FFN layers in ViTs capture class-related information, and both the neuron clustering and low-rank decomposition techniques are well-established; thus, the level of novelty is limited.
>
> We thank the reviewer for raising the concern regarding novelty. We would like to clarify the key contributions and the design rationale of our method.
>
> Our work provides a systematic analysis showing that *class-specific* knowledge primarily resides in FFN modules, whereas MHA modules primarily encode *class-agnostic* information. This perspective has not been articulated or leveraged in prior literature, and it directly guides our class-specific model derivation strategy.
>
> Motivated by this insight, Vulcan adopts a **train-then-prune** paradigm, which is central to our contribution. Traditional pruning methods remove weights deemed unimportant, yet those weights may still contain useful knowledge. Consequently, directly pruning them can result in irreversible knowledge loss, especially at higher pruning rates. In contrast, Vulcan first trains the base ViT to create redundancy in the FFN and MHA modules by introducing duplicated neurons in the FFN and encouraging low-rank structures in the MHA. This allows Vulcan to prune these redundant weights with *​nearly no accuracy loss*​.
>
> To introduce such structured redundancy, Vulcan employs **CCNC** and ​**TNNR**​, which are specifically designed for the FFN and MHA modules, respectively. Importantly, these components are not a simple combination of existing techniques but are ​architecturally tailored to ViTs​. For FFN, since FFN neurons exhibit highly interpretable activation patterns, CCNC collapses each neuron cluster onto the neuron with the highest activation, which not only constructs redundant weights but also encourages the model to focus on class-relevant patterns. For MHA, since SVD preserves information optimally under a low-rank constraint, TNNR explicitly introduces low-rank structure into the MHA weights to support lossless SVD. Finally, Vulcan formulates redundancy construction as a constrained optimization problem and solves it using an augmented Lagrangian framework. The entire procedure is coherent and well-motivated.
>
> > **W2**: The application scenario involves only a small number of target classes, which could often be addressed by lightweight models trained on smaller datasets, leaving the necessity of this approach unclear.
>
> Compared with directly fine-tuning a lightweight model, deriving a small model from a large model offers several distinct advantages:
>
> - **Greater deployment flexibility**: Edge devices exhibit highly diverse resource budgets, and it is unrealistic to assume that a suitably sized pretrained lightweight model always exists for every deployment profile. Training a lightweight model from scratch is also undesirable, as poor initialization often leads to suboptimal convergence, as shown in the table below. In contrast, deriving small models from a large base model avoids these restrictions. Vulcan can automatically determines the target architecture and transfer useful knowledge directly from the large model into the derived one.
> - **Higher accuracy**: Large pretrained models possess richer representational capacity and encode high-level knowledge that small models do not contain. Models derived through Vulcan inherit this knowledge and therefore achieve substantially better decision boundaries.
> - **Better scalability in the long term**: The paradigm of deriving small models from a large model is forward-compatible: as foundation models continue to improve, the derived models will naturally become stronger as well, without requiring redesign or retraining from scratch.
>
> To illustrate these points, we derive three small models from DeiT-Base with the same FLOPs as DeiT-Small (using R = 0.73) for the three ImageNet sub-tasks T1/25–T3/25. The table below compares their performance against both pretrained and non-pretrained DeiT-Small fine-tuned on the same tasks:
>
> | Method                     | T1/25 | T2/25 | T3/25 | Avg   |
> |---------------------------|-------|-------|-------|-------|
> | DeiT-Small (w/o init-FT)  | 58.24 | 60.24 | 61.58 | 60.02 |
> | DeiT-Small (w/ init-FT)   | 93.96 | 94.64 | 93.61 | 94.07 |
> | DeiT-Base (Vulcan-0.73)   | 94.80 | 96.08 | 94.65 | 95.18 |
>
> The results support our claims that (i) training a small model from scratch converges to a suboptimal solution, and (ii) small models derived from a large model outperform directly fine-tuned lightweight models.

---

> ### Author Response · Authors · 2025-11-19
> **Response 2/3**
>
> > **W3**: The method section contains an excessive amount of symbols and equations, making the presentation somewhat dense and difficult to follow.
>
> Thank you for the feedback regarding the dense use of symbols and equations in our methods section. We apologize for the difficulty and have taken steps to improve the presentation in the revised paper. In particular, we ensured that all symbols are clearly defined upon first use and added consolidated notation tables (Tables 15 and 16 in Appendix L) for easy reference. We also optimized several formulas to avoid potential misunderstandings, including the activation-value computation in line 214 and the definition of $i^*$ in Equation (4). We hope that these improvements enhance the readability and clarity of the paper while preserving the necessary theoretical rigor.
>
> To further facilitate understanding, we provide a brief roadmap of the equations used in the paper:
>
> - **Determining the target model architecture**. Before post-training, Vulcan must determine the target architecture corresponding to pruning rate $R$. Equations (3), (5), and (6) compute the three pruned dimensions.
> - **Regularization terms**. Vulcan introduces CCNC and TNNR through the regularizers in Equations (4) and (7), which respectively encourage duplicated neurons in FFN modules and low-rank structure in MHA modules.
> - **Overall loss function**. Equation (8) presents the final optimization objective of Vulcan, where the first- and second-order constraints derived from Equations (4) and (7) are combined with the original task loss.
> - **Pruning procedure**. Equations (9)–(11) describe Vulcan’s pruning process after redundancy has been introduced.
>
> We hope this structured explanation helps clarify the mathematical flow of the method. We sincerely appreciate the reviewer’s constructive feedback.
>
> > **Q1**: It is not specified whether the baseline models were also retrained on the same class subsets—could this affect the fairness of the comparisons?
>
> To ensure a fair comparison, we retrained all baseline models on the same sub-task datasets $\mathcal{D_S}$.
>
> > **Q2**: The paper does not compare Vulcan with lightweight models designed for limited-class scenarios; would Vulcan still provide a clear advantage under such settings?
>
> Please see our response to **W2**. In our experiments, the fine-tuned DeiT-Small on the sub-task datasets serves as the lightweight model specifically tailored for limited-class scenarios.

---

> ### Author Response · Authors · 2025-11-19
> **Response 3/3**
>
> > **Q3**: The FFN collapse and MHA low-rank compression losses are directly added together—could there be potential interactions between the two, and is such joint optimization theoretically justified?
>
> To examine whether jointly optimizing the FFN-collapse (CCNC) and MHA low-rank (TNNR) regularizers is reasonable, we conducted a detailed comparison between **joint optimization** ($C+T$) and three alternative optimization schedules:
>
> - $C\rightarrow T$: Apply CCNC-only post-training first; then freeze the FFN modules and apply TNNR-only post-training.
> - $T\rightarrow C$: Apply TNNR-only post-training first; then freeze the MHA modules and apply CCNC-only post-training.
> - $C\leftrightarrow T$: Alternate between CCNC-only and TNNR-only constraints at every step.
>
> For fairness, we ensure that CCNC and TNNR are applied for the same number of steps as in joint optimization. Consequently, the three alternative schedules require **twice** as many total training steps as the joint optimization setting.
>
> We applied all four strategies to derive models from DeiT-Base for three ImageNet sub-tasks (T1/25, T2/25, T3/25) under pruning rate of 0.60. The results are shown in the table below:
>
> | Method               | T1/25 | T2/25 | T3/25 | Avg   |
> |----------------------|-------|-------|-------|-------|
> | $C\rightarrow T$ | 94.88 | 96.40 | 95.37 | 95.55 |
> | $T\rightarrow C$ | 95.04 | 96.16 | 94.73 | 95.31 |
> | $C\leftrightarrow T$ | 95.04 | 96.24 | 95.53 | **95.60** |
> | $C + T$ | 95.04 | 96.24 | 95.53 | **95.60** |
>
> We observe that joint optimization ($C+T$) and alternating optimization ($C\leftrightarrow T$) achieve the best accuracy, but the differences compared with $C\rightarrow T$ and $T\rightarrow C$ are relatively small.  This suggests that the method is robust to the choice of optimization schedule.
>
> To further investigate the potential interactions between CCNC and TNNR, we additionally measured how each regularizer’s value changes when only the *other* regularizer is applied. Specifically, under the $C\rightarrow T$ and $T\rightarrow C$ settings, we recorded the value of the “inactive” loss before and after training with the “active” loss:
>
> | TNNR ($C\rightarrow T$)     | T1/25 | T2/25 | T3/25 | Avg    |
> |--------------|-------|-------|-------|--------|
> | Before CCNC  | 153.05 | 153.05 | 153.05 | 153.05 |
> | After CCNC   | 152.11 | 151.15 | 151.01 | 151.42 |
>
> | CCNC ($T\rightarrow C$)| T1/25  | T2/25  | T3/25  | Avg     |
> |--------------|--------|--------|--------|---------|
> | Before TNNR  | 7145.47 | 7100.57 | 7473.68 | 7239.91 |
> | After TNNR   | 7112.63 | 7062.22 | 7427.46 | 7200.77 |
>
> Interestingly, in both $C\rightarrow T$ and $T\rightarrow C$ settings, the inactive loss consistently decreases during optimization. This indicates that CCNC and TNNR are ​*not conflicting*​; instead, they are ​*mutually reinforcing*​. In other words, optimizing one regularizer naturally moves the model in a direction that is favorable for the other.
>
> Considering this beneficial interaction and the total number of post-training steps, we adopt the joint optimization scheme in Vulcan.
>
> We thank the reviewer again for the constructive feedback. We sincerely hope that our clarifications and additional experiments address your concerns, and we would be very glad to discuss them further. We remain very open to any additional questions or suggestions.

---

> ### Author Response · Authors · 2025-11-26
> **Rebuttal follow-up**
>
> Dear Reviewer QM9G,
>
> I hope this message finds you well. We would like to kindly follow up regarding our rebuttal submission. As the discussion phase will conclude in about a week, we sincerely value any additional feedback you may have. In particular, we would greatly appreciate knowing whether our responses have addressed your concerns or if there are remaining questions that we could further clarify. We would be very glad to provide any additional information promptly.
>
> Thank you again for your time and for your thoughtful reviews.

---

### Official Review · Reviewer_pANN · 2025-11-01

**Soundness:** 3
**Presentation:** 3
**Contribution:** 2
**Rating:** 8
**Confidence:** 2

**Summary:**

The paper targets class-specific edge deployment by compressing general ViTs into compact, per-class models. It hypothesizes FFN layers are more class-specific while MHA is class-agnostic, and proposes a train-then-prune pipeline: CCNC for FFN and TNNR+SVD for MHA under FLOPs/parameter budgets. Across five ViTs, three tasks, and four datasets, the approach keeps only 20–40% of parameters while matching or improving base accuracy, outperforming structured pruning baselines by up to 13.92%, with hardware-friendly dimensional alignment.

**Strengths:**

1. The paper systematically formulates class-specific ViT compression for edge, supports the FFN/MHA functional split with analyses, and pairs CCNC (FFN) with TNNR+SVD (MHA) under explicit budget constraints.
2. Introducing redundancy and then pruning by effective rank/activation clustering mitigates irreversible knowledge loss typical of prune-then-train; results show consistent gains at 0.6/0.8 pruning ratios over strong structured-pruning baselines.
3. Budgets are enforced in GFLOPs/params, post-pruning dimensions are aligned to hardware-friendly multiples (e.g., ×8), and SVD on attention heads is shown competitive with saliency-based pruning.

**Weaknesses:**

1. It could be very interesting if this framework could be extended as a streaming service platform which can publish tuned weights to edge models in real time.
2. Although some hardware cannot support numerical types, the combination of the proposed method and quantization could be important.

**Questions:**

I see you have covered many experimental results in the appendix. I don't have more questions. Good luck.

---

> ### Author Response · Authors · 2025-11-19
> **Response**
>
> We sincerely appreciate the reviewer’s encouraging assessment and recognition of the significance of our work. For a better understanding of our rebuttal and revision, we have summarized your key concerns and our responses as follows:
>
> > **W1**: It could be very interesting if this framework could be extended as a streaming service platform which can publish tuned weights to edge models in real time.
>
> We greatly appreciate this insightful suggestion, and we fully agree that building a streaming platform for real-time delivery of tuned models to edge devices is a valuable and practical direction. In fact, this aligns closely with our planned future work and our long-term goal of enabling end-to-end on-device intelligence at scale.
> We are happy to briefly share our preliminary thoughts. A real-time platform will require maintaining a cloud-side *model zoo* to support highly customized workloads. We envision deploying a collection of lightweight, task-specific models as isolated containerized units, each capable of rapid startup and teardown. With a serverless orchestration mechanism, the system could instantiate the appropriate model container on demand, achieving millisecond- to second-level elasticity while avoiding persistent resource consumption. Based on feedback from edge users, the platform could update existing models or derive new variants from the base model, with a canary-release strategy to ensure service stability. This design naturally supports continuous learning and model reuse, enabling customized and adaptive services as edge conditions change.
>
> > **W2**: Although some hardware cannot support numerical types, the combination of the proposed method and quantization could be important.
>
> We appreciate this insightful comment. As Vulcan is orthogonal to quantization, many edge devices can indeed benefit from integrating our framework with low-bit quantization. In Appendix J.8, we provide experiments combining Vulcan with several model compression techniques, where quantization shows strong potential for further reducing latency and memory footprint.
>
> To more clearly demonstrate this synergy, we additionally report Vulcan’s performance under different pruning rates (R=0.60, 0.80) when paired with FP16 and INT8 quantization on an NVIDIA RTX 4090 (GPU， batch size=256) and an Intel Xeon Platinum 8558 (CPU, batch size=1), respectively. All results are averaged over the models for the nine sub-tasks (T1/25–T9/100) in Table 1.
>
> | Methods (GPU)             | Latency (ms)        | Throughput (image/s) | Memory (GB) | #Param (M) | FLOPs (G) | Acc. (%) |
> |----------------------|----------------------|------------------------|--------------|-------------|-------------|-----------|
> | DeiT-Base            | 274.27              | 933.39                | 2.21         | 86.57       | 17.57       | 81.27     |
> | Vulcan (0.60)        | 136.67 (2.01×)      | 1873.11               | 1.66         | 34.09       | 6.77        | 92.32     |
> | Vulcan + FP16 (0.60) | 43.98 (6.24×)       | 5820.86               | 1.14         | 34.09       | 6.77        | 92.32     |
> | Vulcan (0.80)        | 99.59 (2.75×)       | 2570.61               | 1.40         | 16.69       | 3.26        | 88.22     |
> | Vulcan + FP16 (0.80) | 32.16 (8.53×)       | 7959.18               | 0.99         | 16.69       | 3.26        | 88.22     |
>
> | Methods (CPU)            | Latency (ms)        | Throughput (image/s) | Memory (MB) | #Param (M) | FLOPs (G) | Acc. (%) |
> |----------------------|----------------------|------------------------|--------------|-------------|-------------|-----------|
> | DeiT-Base            | 32.72               | 30.57                 | 139.50      | 86.57       | 17.57       | 81.27     |
> | Vulcan (0.60)        | 21.22 (1.54×)       | 47.12                 | 112.67      | 34.09       | 6.77        | 92.32     |
> | Vulcan+INT8 (0.60)   | 20.65 (1.58×)       | 48.43                 | 57.00       | 34.09       | 6.77        | 88.20     |
> | Vulcan (0.80)        | 15.95 (2.05×)       | 62.72                 | 94.50       | 16.69       | 3.26        | 88.22     |
> | Vulcan+INT8 (0.80)   | 16.19 (2.02×)       | 61.76                 | 81.00       | 16.69       | 3.26        | 84.24     |
>
> These results show that Vulcan maintains stable accuracy while achieving additional improvements in latency, throughput, and memory efficiency when combined with standard low-bit quantization.
>
> We thank the reviewer again for the constructive feedback. We sincerely hope that our clarifications and additional experiments address your concerns, and we would be very glad to discuss them further. We remain very open to any additional questions or suggestions.

---

> ### Author Response · Authors · 2025-11-26
> **Rebuttal follow-up**
>
> Dear Reviewer pANN,
>
> I hope this message finds you well. We would like to kindly follow up regarding our rebuttal submission. As the discussion phase will conclude in about a week, we sincerely value any additional feedback you may have. In particular, we would greatly appreciate knowing whether our responses have addressed your concerns or if there are remaining questions that we could further clarify. We would be very glad to provide any additional information promptly.
>
> Thank you again for your time and for your thoughtful reviews.

---

### Author Response · Authors · 2025-12-01
**Summary Comment for AC**

We sincerely thank you and all reviewers for the time, constructive feedback, and insightful comments. We have carefully addressed every concern raised during the review period and supplemented all necessary analyses and experiments. Our rebuttal provides detailed clarifications, expanded empirical evidence, and improved explanations that we believe resolve the reviewers’ questions and significantly strengthen the paper.

Unfortunately, before the response system closed, we received only one reviewer follow-up, who maintained a positive stance, so we were unable to engage in further discussion with the remaining reviewers. To assist you in assessing our work, we summarize the key issues we addressed below.

- **Motivation & necessity clarified**: Regarding the concern that our application scenario could be handled by directly fine-tuning lightweight models, we demonstrate that deriving compact models from a large base model offers clear advantages in deployment flexibility, accuracy, and long-term scalability. Empirically, models derived with Vulcan substantially outperform both pretrained and non-pretrained small models fine-tuned on the same tasks.
- **Method presentation streamlined**: We improved the clarity of the method by reorganizing the notation, consolidating all symbols into notation tables, and refining ambiguous formulas. In the rebuttal, we additionally provided a structured roadmap that clarifies how each group of equations contributes to architecture selection, regularizer definition, objective formulation, and pruning, which we will incorporate into the revised paper. These revisions substantially enhance readability without sacrificing rigor.
- ​**Regularizer interactions validated**​: To address the question of whether the CCNC and TNNR regularizers interact undesirably, we conducted a new study comparing joint optimization with multiple sequential and alternating schedules. The results show that all schedules achieve similar performance, and notably, the two regularizers are mutually reinforcing rather than conflicting—supporting the theoretical soundness of our joint optimization design.
- **Computational overhead quantified**: We addressed concerns about computational overhead by detailing our efficiency-oriented implementation choices, measuring wall-clock cost across sub-tasks, pruning rates, and model scales, and providing end-to-end training time and memory comparisons on multiple hardware platforms. The results show that Vulcan introduces modest overhead (e.g., \~0.56s per step on A40 for DeiT-Base), scales sub-linearly with model size, and remains practical even for large models.
- **Novelty and design rationale strengthened**: Regarding novelty, we clarified that our contributions are twofold: (i) we provide new empirical evidence revealing the distinct distribution of class-specific and class-agnostic knowledge across FFN and MHA modules in ViTs, and (ii) we propose a new train-then-prune paradigm that explicitly constructs structured redundancy before pruning. Building on this paradigm, we design CCNC and TNNR as module-tailored regularizers for FFN and MHA, respectively, and integrate them into a coherent augmented-Lagrangian framework to enable near-lossless pruning, substantially beyond a simple combination of existing techniques.

We once again thank the reviewers for their valuable feedback. The rebuttal process has significantly strengthened both our paper and our understanding, and we hope this concise summary helps you efficiently assess our contributions and the substantial effort we invested in addressing all concerns.

---

### Meta-Review · Area_Chair_BuDo · 2025-12-30

**Summary:**

The reviewer's weaknesses are mostly about on novelty, necessity relative to smaller models, and method presentation. Reviewer QM9G argues that the idea “builds on the known observation that FFN layers in ViTs capture class-related information” and that neuron clustering and low-rank decomposition are well established, and questions whether “lightweight models trained on smaller datasets” could address the scenario, also finding the method section “dense and difficult to follow.” Reviewer Z9iw similarly notes that the writeup is “somewhat complicated,” that the method section is “dense and overly mathematical,” and feels the methodology “more like a collection of engineering techniques” than a focused exploration. Reviewer J15G asks for “more rigorous verification” of the MHA class-agnostic claim on other models/datasets, requests overhead data, and suggests including a sketch of the theory in the main paper. Reviewer pANN does not raise major methodological weaknesses, but suggests interesting extensions to streaming service platforms and to combining with quantization.

The AC recommends acceptance, following reviewer's consensus, because the paper tackles a meaningful edge‑deployment setting with a coherent, empirically validated framework that goes beyond straightforward pruning or small‑model finetuning. Vulcan’s key insight—that FFNs and MHAs carry class‑specific and class‑agnostic knowledge differently—and its train‑then‑prune design with CCNC and TNNR lead to class‑specific ViTs that often surpass both the base model and tuned small baselines at 20–40% size, with hardware-aware dimensions and modest training overhead. The authors provide experiments across tasks, architectures, and devices, strengthen their motivation versus lightweight alternatives, clarify their optimization scheme and theoretical underpinnings, and show robustness across random sub‑tasks and imbalance. While some novelty concerns remain about the use of known building blocks, the overall package is practically useful, well‑supported. On balance, AC agrees with positive points raised by all reviewers which outweigh the negative points. The authors are strongly encouraged to include the additional reviewer recommendations, experiments from rebuttal and clarifications in the camera-ready version.

**Reviewer Concerns:**

The rebuttal addresses many concerns in a focused way. For Reviewer QM9G, the authors clarify novelty by framing Vulcan as a new train‑then‑prune paradigm that first constructs structured redundancy tailored to FFN and MHA before pruning, and provide a concrete comparison: DeiT‑Base→Vulcan‑0.73 versus DeiT‑Small with and without pretraining, showing derived models outperform fine‑tuned small models on identical sub‑tasks. They confirm that all baselines are retrained on the same class subsets and add a brief equation roadmap and notation tables to ease reading. For Reviewer J15G, they extend FFN/MHA analyses to multiple model–dataset pairs (DeiT‑Tiny/CIFAR‑10, DeiT‑Small/CIFAR‑100, ViT‑Large/ImageNet, Swin‑T/COCO), showing SVD consistently beating activation/gradient/Taylor pruning in MHA, and they report TNNR overhead (e.g., ~0.56s per step on A40 for DeiT‑Base, sub‑linear scaling with model size, and modest end‑to‑end time/memory increases) plus random sub‑task stability with mean±std at pruning 0.8/0.9. They commit to moving the near‑lossless pruning argument into the main text. For Reviewer Z9iw, they fix ambiguous formulas, add notation tables, provide an intuitive explanation of the augmented Lagrangian, and clarify the coherent design (architecture choice → redundancy‑inducing regularizers → constrained optimization → pruning). They also answer concerns about overfitting and imbalance by showing good behavior on very small sub‑tasks and reporting performance under long‑tailed splits. For Reviewer pANN, they explain how Vulcan could underpin a streaming model‑zoo platform and add experiments combining Vulcan with FP16/INT8 quantization showing further latency and memory gains. These responses substantially reduce the force of the earlier criticisms; some questions of perceived incremental novelty remain, but the practical and empirical contribution is clearer.

**Reviewer Scores:**

Reviewers could not revise scores after the rebuttal, but their follow‑ups suggest likely stability or slight strengthening of positive views. Reviewer pANN already gives “8: accept, good paper (poster)” and had no additional questions beyond future directions. Reviewer J15G explicitly says “I tend to keep my positive score” after reading the rebuttal. Reviewer Z9iw did not respond again, but their main issues were about clarity and coherence, which are directly addressed. Reviewer QM9G, at “4: marginally below the acceptance threshold. But would not mind if paper is accepted,” may still see novelty as moderate, but the added comparisons to small models and the demonstrated overhead profiles address the core of their W2 and W3, so a modest upward / increase of the score would be plausible.

---

### Decision · Program_Chairs · 2026-01-26

Accept (Poster)